# The Impact of Scaling Training Data on Adversarial Robustness

**Marco Zimmerli**
ETH Zürich
mzimmerli@ethz.ch

**Andreas Plesner**
ETH Zürich
aplesner@ethz.ch

**Till Aczel**
ETH Zürich
taczel@ethz.ch

**Roger Wattenhofer**
ETH Zürich
wattenhofer@ethz.ch

## Abstract

Deep neural networks remain vulnerable to adversarial examples despite advances in architectures and training paradigms. We investigate how training data characteristics affect adversarial robustness across 36 state-of-the-art vision models spanning supervised, self-supervised, and contrastive learning approaches, trained on datasets from 1.2M to 22B images. Models were evaluated under six black-box attack categories: random perturbations, two types of geometric masks, COCO object manipulations, ImageNet-C corruptions, and ImageNet-R style shifts. Robustness follows a logarithmic scaling law with both data volume and model size: a tenfold increase in data reduces attack success rate (ASR) on average by 3.2%, whereas a tenfold increase in model size reduces ASR on average by 13.4%. Notably, some self-supervised models trained on curated datasets, such as DINOv2, outperform others trained on much larger but less curated datasets, challenging the assumption that scale alone drives robustness. Adversarial fine-tuning of ResNet50s improves generalization across structural variations but not across color distributions. Human evaluation reveals persistent gaps between human and machine vision. These results show that while scaling improves robustness, data quality, architecture, and training objectives play a more decisive role than raw scale in achieving broad-spectrum adversarial resilience.

## 1 Introduction

Deep neural networks have achieved remarkable success in computer vision tasks [31, 44, 32, 20, 40]. Yet, their vulnerability to adversarial examples remains a fundamental challenge to their deployment in safety-critical applications [48]. Adversarial examples are inputs with semantic preserving changes that cause misclassifications, revealing a significant gap between human and machine perception [17]. While humans recognize objects under various distortions, state-of-the-art models can be fooled by imperceptible modifications [48, 38]. This vulnerability raises profound questions about the nature of learned representations and the factors that determine model robustness.

The asymmetry between human and machine perception creates critical security vulnerabilities across deployed systems. In visual domains, adversarial perturbations that remain imperceptible or semantically clear to humans can cause catastrophic failures in machine vision applications. Content moderation systems exemplify this vulnerability, where malicious actors can craft adversarial examples to evade automated filters while the harmful content remains readily identifiable to human observers [1, 46]. Similarly, autonomous vehicle perception systems can be manipulated through

39th Conference on Neural Information Processing Systems (NeurIPS 2025) Workshop: Reliable ML from Unreliable Data.

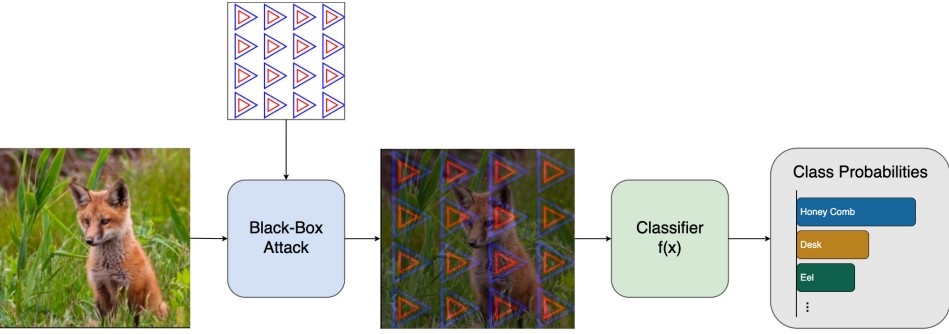

Figure 1: Overview of the black-box attack pipeline. Input images are modified using a semantic adversarial attack. In this example, an ImageNet image of a red fox is attacked using the Geometric-MasksV2 3-4-2 C1 with an opacity of 128, causing a misclassification in the target classifier.

carefully designed perturbations that preserve semantic meaning for human drivers but induce misclassifications in computer vision models [8]. These vulnerabilities extend beyond visual tasks, as analogous techniques compromise malware detection and other pattern recognition systems [18, 5]. The fundamental gap in robustness between biological and artificial perception thus constitutes a systematic attack surface rather than merely a theoretical limitation.

Recent advances in vision model architectures have produced increasingly sophisticated systems, from Vision Transformers [14] to self-supervised models like DINOv2 [37] and multi-modal architectures such as CLIP [41]. These models employ fundamentally different training paradigms, namely supervised, self-supervised, and contrastive learning, and are trained on datasets of unprecedented scale that range from millions to billions of images. While conventional wisdom suggests that larger datasets and more sophisticated training objectives should confer greater robustness, empirical evidence reveals a more nuanced reality. The interplay between data quantity, curation quality, and training paradigm produces unexpected robustness patterns, with some smaller, carefully curated datasets yielding models more robust than those trained on orders of magnitude more data [12, 27]. Despite extensive research on individual factors, the relationship between these training characteristics and resulting adversarial robustness remains poorly understood.

This work investigates how training data shapes adversarial robustness in vision models. We systematically evaluate 36 state-of-the-art image classification models across six distinct black-box attack categories, ranging from simple color perturbations to complex geometric occlusions and artistic domain shifts. Our analysis spans models trained on datasets from 1.2 million to 22 billion images, enabling insights into scaling laws for adversarial robustness. An overview of our attack pipeline can be found in Figure 1. We address four central research questions:

1. Does training data scale influence model vulnerability to different adversarial perturbations?
2. Do self-supervised and contrastive learning paradigms inherently produce more robust representations than supervised training?
3. Despite targeted adversarial training, can novel geometric mask configurations always be constructed to exploit vulnerabilities in fine-tuned models?
4. Can adversarial fine-tuning align model robustness with human perceptual invariance?

Our investigation employs a comprehensive black-box evaluation framework that emphasizes semantic validity over bounded perturbations [38]. Unlike traditional $\ell_p$-norm constrained attacks [17, 36], we focus on perturbations that preserve semantic content while exploiting model vulnerabilities, such as geometric masks [30], artistic renditions [23], and naturalistic corruptions [22].

Our work makes four principal contributions to understanding adversarial robustness. First, we establish quantitative scaling laws demonstrating that robustness improvements saturate logarithmically with training data volume. Additionally, attack-specific variations reveal fundamental differences in how models handle spatial versus stylistic perturbations. We further find that scale without quality control offers minimal benefits for CLIP models, indicating that strategic data curation supersedes volume for comprehensive adversarial robustness. Second, we show that the training paradigm has

minimal impact on robustness compared to data curation quality and model scale. Third, through targeted adversarial fine-tuning experiments with geometric masks, we demonstrate that ResNet50 models can learn robust features that generalize across structural variations but fail to transfer across color distributions. Fourth, our human evaluation studies reveal persistent gaps between human and machine vision, with even the best models exhibiting vulnerabilities that humans navigate effortlessly.

## 2 Related Work

**Adversarial Examples and Attack Methods**   The vulnerability of neural networks to adversarial examples has been a central concern since their discovery [48, 5]. These imperceptible perturbations, which cause misclassifications, have driven the development of increasingly effective attacks. The Fast Gradient Sign Method (FGSM) introduced an efficient single-step gradient ascent approach [17], followed by iterative methods such as Projected Gradient Descent (PGD) for stronger attacks [36]. The AutoAttack framework unified multiple complementary attacks into a parameter-free benchmark for reliable, reproducible evaluation [10]. It is also the default evaluation method in RobustBench, which ranks models for consistent, reproducible robustness comparisons [11].

Beyond small $\ell_p$-bounded perturbations, researchers have explored semantic adversarial examples that preserve image meaning while drastically altering model predictions. State-of-the-art models turn out to be vulnerable to comparably natural classes of perturbations like translations and rotations [15]. Research on semantic adversarial examples introduced HSV color space transformations, demonstrating that shifting hue and saturation components while preserving brightness can reduce CNN accuracy to below 10% on CIFAR-10 [25]. This approach exploits the shape bias of human vision, generating naturally appearing images that contain the original object with different colors. Recent work on Generative Adversarial Training (GAT) and composite adversarial attacks builds on these ideas by integrating multiple semantic perturbations, such as hue, saturation, brightness, contrast, and rotation, to construct more comprehensive threat models [28].

**Defense Mechanisms**   Adversarial training has emerged as the predominant defense mechanism due to its conceptual simplicity and empirical effectiveness [36, 17]. This approach incorporates adversarially perturbed examples during training to improve model robustness. Recent advances have demonstrated that adversarial training benefits substantially from increased training data volume, exceptionally high-quality synthetic data [42].

**Data Distribution and Robustness Sensitivity**   A critical finding in adversarial robustness research concerns the sensitivity of robust accuracy to input data distributions. It has been demonstrated that semantically-preserving transformations of data distributions can drastically alter the adversarial robustness of models, even when retrained on the transformed distribution [12].

**Scaling Trends in Adversarial Robustness**   CIFAR-10 adversarial robustness has been studied by training WideResNets with large synthetic datasets and evaluating under white-box conditions using AutoAttack and 40-step PGD on the models' CW loss, showing that robustness scales with model and dataset size but plateaus near 90% accuracy, partly due to invalid adversarial images that also fool humans[3]. Recent investigations into scaling laws for adversarial robustness of language models reveal that, unlike standard accuracy, larger language models do not consistently exhibit improved robustness [27]. In the same line of work, offense-defense balance analyses indicate that increasing attack compute currently outpaces defense improvements for fixed model sizes. However, larger models exhibit more favorable defense scaling properties, hinting that scaling model capacity may eventually shift the advantage toward defense [27]. Further, it has been shown that larger models generally achieve higher $\ell_\infty$-robust accuracy under the AutoAttack on ImageNet [47].

**Synthetic Data Quality and Training Efficiency**   The quality of synthetic training data, often measured through Fréchet Inception Distance (FID) [24], has a significant impact on adversarial robustness outcomes [3]. Recent work has incorporated data quality metrics into scaling laws, demonstrating that higher-quality synthetic data enables more compute-efficient adversarial training. In contrast, low-quality data limits the benefits of scaling [3].

# 3 Experimental Design for Robustness Scaling Analysis

## 3.1 Model Setup

The analysis was carried out on ViT [14], ResNet [19], CLIP [41], DINOv1 [7], DINOv2 [37], Swin [33], Swinv2 [34], ConvNeXt [35], YOLO [50], ViT-MAE [21], PaliGemma [4], BEiT [2], BEiTv2 [39], SigLIP [51] and SigLIPv2 [49] models, where the exact specifications can be found in the Appendix Table 2. All models in this study were evaluated on the ImageNet-1K classification task (validation split) for benchmarking image recognition [45]. Each model utilized its standard preprocessing transformations during evaluation.

CLIP models were evaluated in a zero-shot classification setting without additional training, using prompts of the form "a photo of a {class name}" for each of the 1,000 ImageNet classes, following standard CLIP evaluation protocols [41]. All variants use the OpenCLIP framework [9], except for the Apple DFN CLIP models [16], which were loaded from the Hugging Face model repository.

DINOv1, ViT-MAE, and PaliGemma were not initially designed for ImageNet classification, so a single linear classification head was appended to the frozen backbone features. The backbone was frozen while only the classification head was trained on the ImageNet training split using Cross-Entropy Loss. The exact training specifications can be found in the Appendix Table 3.

## 3.2 Evaluation Metrics

Following Dong et al. [13], we evaluate models using accuracy and attack success rate (ASR). Let $\mathcal{D} = \{(x_i, y_i)\}_{i=1}^{N}$ be a dataset of $N$ image-label pairs, where $x_i$ is an input sample and $y_i$ its corresponding ground-truth label. We write $C(\cdot)$ for a classifier and $A(\cdot)$ for an adversarial attack.

**Accuracy** The accuracy of a classifier $C$ on a dataset $\mathcal{D}$ is defined as

$$\text{Acc}(C, \mathcal{D}) = \frac{1}{|\mathcal{D}|} \sum_{(x,y) \in \mathcal{D}} \mathbf{1}[C(x) = y], \quad \mathbf{1}[\cdot] \text{ is the indicator function.}$$

**Attack Success Rate (ASR)** ASR is defined as the ratio of initially correctly classified images that become misclassified after applying the adversarial attack. Let $\mathcal{S}_{\text{correct}} = \{(x, y) \in \mathcal{D}_{\text{clean}} \mid C(x) = y\}$ be the subset of correctly classified images by classifier $C$ from the clean dataset $\mathcal{D}_{\text{clean}}$. Then

$$\text{ASR}(C, A, \mathcal{S}_{\text{correct}}) = \frac{1}{|\mathcal{S}_{\text{correct}}|} \sum_{(x,y) \in \mathcal{S}_{\text{correct}}} \mathbf{1}[C(A(x)) \neq y].$$

**Approximation of Attack Success Rate** For certain attack scenarios where only the adversarially attacked dataset $\mathcal{D}_{\text{adv}} = \{(A(x_i), y_i)\}_{i=1}^{N}$ is available, we approximate the ASR using a surrogate clean dataset $\mathcal{D}_{\text{surrogate}} = \{(x_i^{\text{sur}}, y_i^{\text{sur}})\}_{i=1}^{M}$ that resembles the unavailable original $\mathcal{D}_{\text{clean}}$. The approximated ASR is computed as

$$\text{ASR}_{\text{approx}}(C, \mathcal{D}_{\text{adv}}, \mathcal{D}_{\text{surrogate}}) = \frac{\text{Acc}(C, \mathcal{D}_{\text{surrogate}}) - \text{Acc}(C, \mathcal{D}_{\text{adv}})}{\text{Acc}(C, \mathcal{D}_{\text{surrogate}})}.$$

To validate this approximation, we evaluated all 36 models on all 19 attacks for which both clean and adversarial images are available, and compared the resulting approximated and actual ASR values. The approximation systematically underestimated the actual ASR by an average of 3.09%-points ($\sigma = 1.93$%-points), computed over all 684 evaluation points (36 models × 19 attacks). This underestimation occurs because the approximation cannot account for initially misclassified samples that become correctly classified under adversarial perturbation, a phenomenon that reduces the apparent accuracy drop. Despite this bias, the approximation provides consistent relative rankings across models, enabling meaningful comparative analysis when actual ASR computation is infeasible.

## 3.3 Attacks

We consider a black-box threat model in which adversaries have no access to the model's architecture, parameters, or gradients [38]. Moreover, we impose no constraints on the perturbation magnitude,

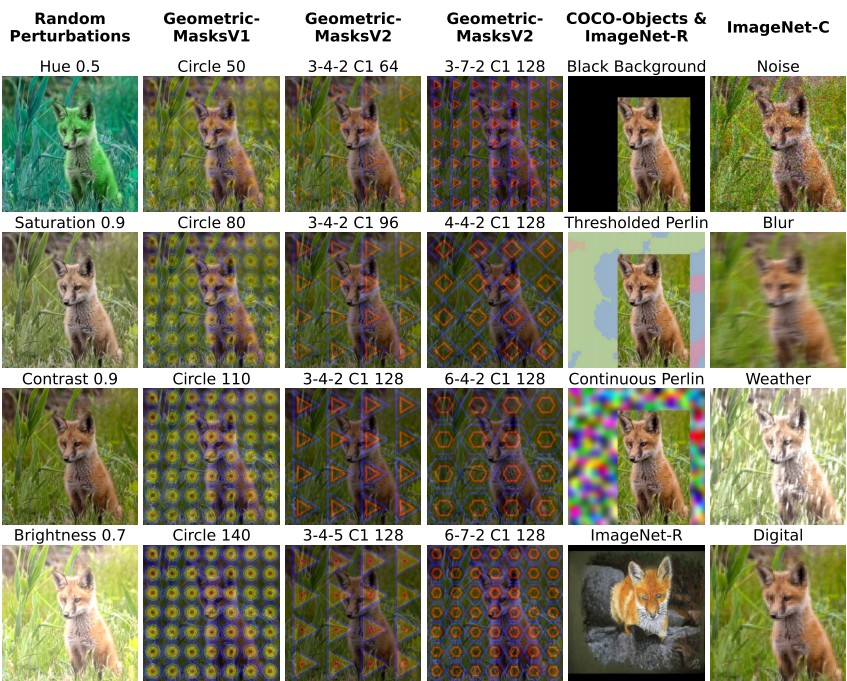

Figure 2: Sample images of all attacks applied for the robustness analysis of the categories Random Perturbations, GeometricMasksV1, GeometricMasksV2, COCO-Objects, ImageNet-R, and ImageNet-C. The original image is from the ImageNet class red fox.

thereby enabling exploration of a broader range of attack strategies, including those that introduce perceptible yet semantically consistent perturbations [30]. Figure 2 shows a sample for every attack applied in the robustness analysis, while further sample images are available in Appendix C. We provide descriptions of the attacks in Appendix B.

# 4   Adversarial Fine-Tuning

To investigate the relationship between adversarial training and model generalization capabilities, we fine-tuned three ResNet50 models using different GeometricMasksV2 configurations. These experiments assess whether models can learn robust features from structured adversarial examples and generalize beyond the specific perturbations encountered during training. Each model was initialized from ImageNet pre-trained weights and fine-tuned on a modified version of the ImageNet training set, where a controlled percentage of images were augmented using the GeometricMasksV2 attack. The fine-tuning process targeted all model parameters while maintaining the original architecture.

Following fine-tuning, the models underwent evaluation on various GeometricMasksV2-based adversarial attacks, including a novel color scheme, C3, and a 45-degree rotated mask, named C4. Examples of all the GeometricMasksV2 applied in this evaluation can be found in Figure 13 with more in Appendix D.

Table 1: Fine-tuning configurations for ResNet50 models with GeometricMasksV2 augmentation. All models were trained with a batch size of 64 for three epochs.

| Model Variant | Mask Type | Color Scheme | Opacity | Adversarial Examples |
|---|---|---|---|---|
| ResNet50-v1 | 3-4-2 | C1 | 64 | 50% |
| ResNet50-v2 | 3-4-2 | C1 & C2 | 64 | 50% |
| ResNet50-v3 | Random | C1 & C2 | 64 | 50% |

# 5  Human-Model Alignment

The human evaluation employed the GeometricMasksV2 attack with configuration 6-7-2 C1, identified in our robustness analysis as producing the highest attack success rates for some of the evaluated models. This evaluation serves not only to assess the attack's effectiveness on humans but also to confirm that it continues to produce valid semantic adversarial examples. The geometric mask was applied to the ImageNette dataset, a curated subset of ImageNet comprising ten visually distinct classes [26]. It provides a simplified classification task well-suited for human participants. The evaluation protocol consisted of:

- **Dataset**: 25 randomly selected ImageNette images per difficulty level
- **Task**: 10-way classification among ImageNette categories without time constraints
- **Perturbation**: GeometricMasksV2 (6-7-2 C1)
- **Difficulty** levels:
    - Baseline: Opacity 0 (no occlusion)
    - Easy: Opacity 64 (minimal occlusion)
    - Medium: Opacity 96 (moderate occlusion)
    - Hard: Opacity 128 (substantial occlusion)

The graphical user interface employed in the evaluation is illustrated in Appendix Figure 15. A total of six human participants completed the evaluation protocol, and we report the mean accuracy for each difficulty level across these individuals. In parallel, five models were evaluated on the complete ImageNette dataset, including the adversarially fine-tuned ResNet50-v1 described in Section 4. For a fair comparison with human participants, model predictions were restricted to the ten ImageNette classes. This experimental design enables a direct comparison between human and model performance across difficulty levels. The results confirm the validity of the adversarial examples as human participants consistently achieved accuracies exceeding 93% at all difficulty levels. While the adversarially fine-tuned ResNet50-v1 showed substantially improved robustness, the other model's performance often deteriorated significantly starting at the easy level.

# 6  Results

## 6.1  Robustness Scaling Analysis

Due to space, some results are moved to Appendix G. For instance, results show that contrastive learning shows the lowest average ASR at 27.9%, while self-supervised learning is at 28.4%, and supervised learning has the highest vulnerability with 34.3% ASR on average.

### 6.1.1  Datasets and Adversarial Robustness

Figure 3 presents the comprehensive robustness evaluation across all attack categories using the overall average attack success rate. This metric is calculated as the mean of the average ASR values for each attack category: Random Perturbations, GeometricMasksV1, GeometricMasksV2, COCO Objects, ImageNet-C, and ImageNet-R. It provides a holistic assessment of model vulnerability. Lower ASR values indicate superior robustness across diverse adversarial conditions.

The relationship between training data scale and overall robustness follows a logarithmic function: $ASR = -3.16 \log_{10}(x) + 55.53$, where $x$ represents the training dataset size in number of images. This enhanced scaling coefficient suggests that increased training data provides cumulative benefits across multiple robustness dimensions rather than specialized defenses against specific perturbations. Note that this scaling law does not account for the correlation of 0.59 between dataset size and model size. See Section 6.1.3 for a scaling law with separated training data size and model size.

CLIP models do not follow the scaling trend as closely as expected. Despite some top performers like CLIP-EVA02-L-14, most models underperform given their training data scale, especially the smallest architecture, CLIP-ViT-B-32. The degraded performance presumably occurs due to the lower training data quality in the web-collected image-text pairs. This implies that scale without quality control offers minimal benefits, indicating that strategic data curation supersedes volume for comprehensive

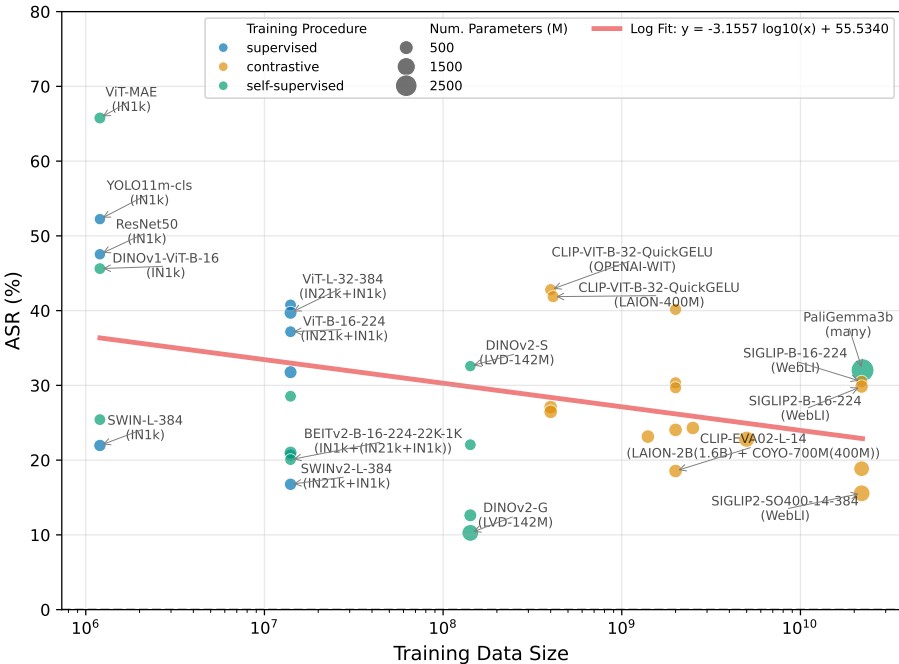

Figure 3: Overall average ASR across different attack categories: Random Perturbations, Geometric-MasksV1, GeometricMasksV2, COCO Objects, ImageNet-C, and ImageNet-R. The overall average is computed as the mean of the average ASR values for each attack category. We show in Figure 16 the same plot with labels for all points.

adversarial robustness. Notably, for a given CLIP architecture, models almost always exhibit superior robustness when trained on larger datasets. In Appendix F.1, we provide a detailed comparison of four CLIP-ViT-L-14 models.

Self-supervised DINOv2 models dominate the low-ASR regime. DINOv2-G achieves the lowest overall ASR at 10.3%, closely followed by DINOv2-L, establishing a clear performance hierarchy within the DINOv2 family that correlates with model scale. This consistent scaling behavior, also observed for other architectures, indicates that an increase in model size can improve the robustness within a model family.

Web-scale trained models occupy the second tier of performance. The SigLIP-SO400 and SigLIP2-SO400 variants, trained on WebLI with roughly 155 times more data than DINOv2, achieve ASRs that nearly match those of DINOv2-G despite employing fundamentally different training paradigms.

Traditional supervised models show a significant variance in the vulnerability across the evaluation suite. ResNet50 and YOLO11m-cls occupy the high-ASR region. Swin-L-384 achieves a comparable ASR despite only training on ImageNet-1K. Swinv2-L-384 further improves with an updated architecture and additional training on ImageNet-21K, achieving a superb ASR of 16.8%. The performance of the Swin models indicates that the architecture and the training procedure may compensate for their small training datasets.

### 6.1.2 Model Scale and Adversarial Robustness

We find the relationship between model size and adversarial robustness reveals a consistent scaling law as shown in Figure 4, with attack success rates decreasing logarithmically as model parameters increase. Larger models exhibit significantly reduced vulnerability to adversarial perturbations, following the relationship $ASR = -13.39 \log_{10}(x) + 141.18$. This robust scaling behavior spans multiple orders of magnitude, from compact models like ResNet50 with high ASRs approaching 50%, to massive architectures such as DINOv2-G achieving ASRs below 15%. The logarithmic nature of this relationship suggests small returns as model scale increases. Yet, the consistent downward trend across diverse architectures and training paradigms indicates that increased parameter count provides a fundamental defensive advantage against adversarial attacks. Notably, this size-robustness

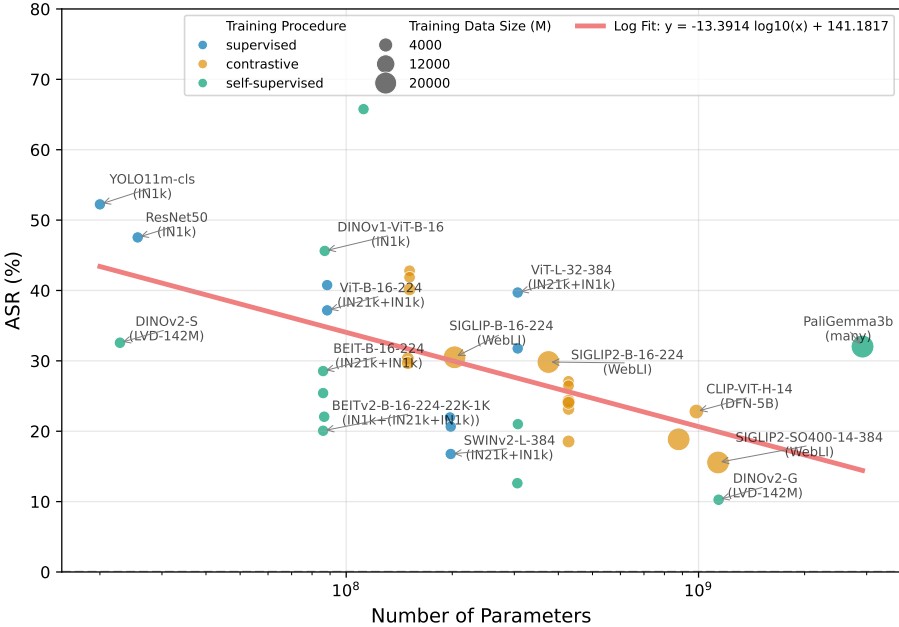

Figure 4: Overall average ASR relative to the number of model parameters averaged across: Random Perturbation, GeometricMasksV1, GeometricMasksV2, Coco Objects, ImageNet-C, and ImageNet-R attacks. We show in Figure 17 the same plot with labels for all points.

correlation appears largely independent of training methodology, as models of similar scale cluster together despite employing different learning objectives, suggesting that the sheer capacity to learn complex representations may be more critical for adversarial robustness than the specific training approach.

### 6.1.3 Fitting a Two-Variable Scaling Law for ASR

The univariate scaling laws presented above do not consider the correlation between training dataset size and model size, which in our data is 0.59. This correlation reflects the common practice that larger models are typically trained on larger datasets. Ignoring this relationship can bias the interpretation of how each factor independently influences attack success rate (ASR).

To account for the joint effect of dataset and model size, we computed a bivariate scaling law. To make this relationship separable, so that the contributions of training dataset size and model size can be individually assessed, we applied principal component analysis (PCA) to the log-transformed data. We then fitted a linear function in the PCA space and projected the result back to the original variables. The resulting bivariate scaling law is:

$$\text{ASR} = -0.46 \log_{10}(x_{data}) - 12.53 \log_{10}(x_{model}) + 137.67$$

where $x_{data}$ is the training dataset size and $x_{model}$ is the model size. This result indicates that model size has a more pronounced effect on ASR than dataset size. Both univariate and bivariate approaches have their validity. While the implicit correlation can influence the univariate scaling law, since larger models tend to be trained on larger datasets, the bivariate law allows us to separate these effects. Nevertheless, in practice, one cannot train huge models on small datasets and vice versa, so the univariate scaling law still provides relevant insights in realistic training regimes.

While the model size shows better scaling, the data scaling can be beneficial as it does not impact inference costs.

### 6.2 Human-Model Alignment

Figure 5 presents the comparative evaluation of human and model performance on ImageNette under GeometricMasksV2 (6-7-2 C1) perturbations at varying opacity levels. The baseline (opacity 0) represents unperturbed ImageNette images, where all achieve near-perfect accuracy.

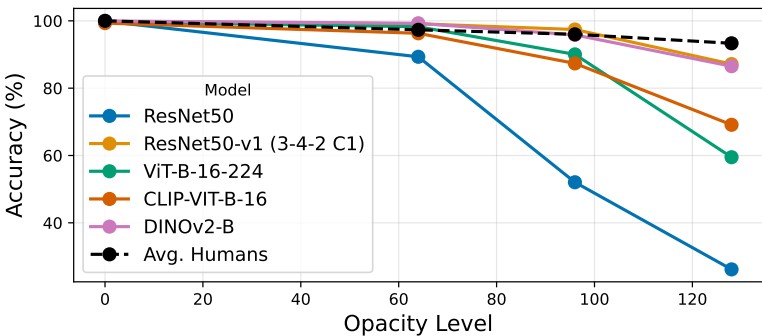

Figure 5: Accuracies of various models and the average accuracy of human participants on the GeometricMasksV2 6-7-2 C1 mask, applied at opacities 0, 64, 96, and 128. Raw values in Table 6.

Human participants demonstrate superior robustness across all opacity levels. The gradual degradation of the human participants' accuracy contrasts sharply with the steeper performance declines observed in computational models, showing that model limitations cause the performance drops.

The fine-tuned ResNet50 (ResNet-v1) and DINOv2-B perform the closest to humans. The models demonstrate that appropriate training strategies, whether through adversarial fine-tuning or self-supervised learning, can substantially enhance robustness to geometric perturbations.

The models ViT-B-16-224 and CLIP-VIT-B-16 display similar performance profiles through moderate perturbation levels, but drop significantly after opacity 64. This performance cliff indicates a fundamental limitation in handling severe geometric occlusions despite robustness to mild perturbations.

The vanilla ResNet50 demonstrates pronounced vulnerability even at minimal perturbation levels. This performance pattern underscores the critical importance of specialized training, as the identical architecture achieves near-human robustness when fine-tuned with geometric masks.

These results establish that humans are more robust than models. Further, the results demonstrate that the mask 6-7-2 C1, the most severe attack in GeometricMasksV2, renders valid adversarial examples even at opacity 128. The persistent gap between human and model performance, particularly at high opacity levels, reveals a fundamental vulnerability that can be exploited in adversarial settings. At opacity 128, even the best-performing models misclassified approximately 13% of images that humans correctly identify, demonstrating that carefully crafted perturbations can selectively impair machine vision while preserving human interpretability. This asymmetry highlights the differences in robustness mechanisms between biological and artificial vision systems.

# 7 Conclusion

Our evaluation of 36 vision models reveals that adversarial robustness is governed by clear, logarithmic scaling laws concerning model *and* dataset size. The relationship for training data is $ASR = -3.1557 \log_{10}(x) + 55.5340$, and for model size is $ASR = -13.3914 \log_{10}(x) + 141.1817$. However, scale is not the sole determinant of resilience. The superior performance of models like DINOv2, trained on highly curated data, indicates that quality can be more impactful than sheer volume. We found that the training paradigm—supervised, self-supervised, or contrastive—has a limited effect on robustness, suggesting architectural and data characteristics are more critical.

Adversarial fine-tuning on geometric masks confirmed that models can learn to generalize across structural variations like shape, scale, and rotation. However, this robustness is brittle and fails to transfer to unseen color schemes, indicating that geometric and chromatic invariance are learned separately. Furthermore, human evaluators consistently outperformed all models, including fine-tuned ones, highlighting a persistent and fundamental gap between biological and artificial visual systems.

Our study is limited by the lack of standardized dataset documentation and a focus on black-box attacks. Future work should expand the attack taxonomy to include gradient-based methods to test if these scaling trends hold. Extending evaluations to tasks like object detection and segmentation would further clarify how data and model scale influence robustness in scenarios requiring complex spatial reasoning.

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

# A  Model Specifications of the Robustness Scaling Analysis

# B  Attacks

As mentioned in the main text, we consider a black-box threat model in which adversaries have no access to the model's architecture, parameters, or gradients [38], and we impose no constraints on the perturbation magnitude, enabling exploration of a broader range of attack strategies. Appendix C contains additional sample images.

## B.1  Random Perturbations

The Random Perturbations comprise four distinct perturbation variants, each targeting a specific image property: hue, saturation, contrast, or brightness. These attacks modify their respective properties within predefined bounds through uniform random sampling for each image. The attacks are implemented as dynamic transformations integrated directly into the model's preprocessing pipeline using the Python library Kornia [43]. The perturbations are applied before any standard preprocessing operations, ensuring that the model encounters perturbed inputs without any prior adaptation. Each variant operates independently and was applied to the entire ImageNet validation split.

## B.2  GeometricMasksV1

The GeometricMasksV1 attack employs HCaptcha-inspired geometric overlays to evaluate model robustness against structured occlusions [30]. This attack category comprises four distinct mask patterns, namely Circle, Diamond, Square, and Knit, each designed to systematically obscure portions of the input image while preserving overall semantic interpretability. We applied the Circle mask to the entire ImageNet validation set at four opacity levels: $\alpha \in \{50, 80, 110, 140\}$, where $\alpha$ represents the opacity value on a 0-255 scale.

## B.3  GeometricMasksV2

GeometricMasksV2 extends the functionality of GeometricMasksV1 by enabling more flexible parametrization of mask configurations [29]. The naming convention follows a systematic format:

Table 2: Models used in the Robustness Scaling Analysis. IN1k = ImageNet-1K, IN21k = ImageNet-21K. A bracket following the dataset name (e.g., LAION-2B(1.6B)) indicates the size of the subset (1.6B images) used during training.

| Model Name | Training Dataset | Training Procedure | Num. Parameters (M) | Training Data Size (M) |
|---|---|---|---|---|
| ResNet50 | IN1k | supervised | 25.6 | 1.2 |
| ViT-B-16-224 | IN21k+IN1k | supervised | 88.3 | 14.0 |
| ViT-B-32-384 | IN21k+IN1k | supervised | 88.3 | 14.0 |
| ViT-L-16-384 | IN21k+IN1k | supervised | 306.7 | 14.0 |
| ViT-L-32-384 | IN21k+IN1k | supervised | 306.7 | 14.0 |
| CLIP-VIT-B-16 | DFN-2B | contrastive | 149.6 | 2000.0 |
| CLIP-VIT-B-32-QuickGELU | OPENAI-WIT | contrastive | 151.3 | 400.0 |
| CLIP-VIT-B-32-QuickGELU | LAION-400M | contrastive | 151.3 | 413.0 |
| CLIP-VIT-B-32 | LAION-2B | contrastive | 151.3 | 2000.0 |
| CLIP-EVA02-B-16 | LAION-2B(1.6B) + COYO-700M(400M) | contrastive | 149.6 | 2000.0 |
| CLIP-VIT-L-14-QuickGELU | MetaClip400M | contrastive | 427.6 | 400.0 |
| CLIP-VIT-L-14-QuickGELU | OPENAI-WIT | contrastive | 427.6 | 400.0 |
| CLIP-VIT-L-14 | DataComp-1B | contrastive | 427.6 | 1400.0 |
| CLIP-VIT-L-14 | MetaClip full CC | contrastive | 427.6 | 2500.0 |
| CLIP-VIT-L-14 | DFN-2B | contrastive | 427.6 | 2000.0 |
| CLIP-EVA02-L-14 | LAION-2B(1.6B) + COYO-700M(400M) | contrastive | 427.6 | 2000.0 |
| CLIP-VIT-H-14 | DFN-5B | contrastive | 986.1 | 5000.0 |
| DINOv2-S | LVD-142M | self-supervised | 22.8 | 142.0 |
| DINOv2-B | LVD-142M | self-supervised | 86.6 | 142.0 |
| DINOv2-L | LVD-142M | self-supervised | 306.0 | 142.0 |
| DINOv2-G | LVD-142M | self-supervised | 1140.0 | 142.0 |
| SWIN-L-384 | IN1k | supervised | 197.0 | 1.2 |
| SWINv2-L-384 | IN21k+IN1k | supervised | 198.0 | 14.0 |
| ConvNext-L | IN21k+IN1k | supervised | 198.0 | 14.0 |
| YOLO11m-cls | IN1k | supervised | 20.0 | 1.2 |
| ViT-MAE | IN1k | self-supervised | 112.0 | 1.2 |
| DINOv1-ViT-B-16 | IN1k | self-supervised | 86.9 | 1.2 |
| PaliGemma3b | many | self-supervised | 2920.0 | 22256.0 |
| BEIT-B-16-224 | IN21k+IN1k | self-supervised | 86.0 | 14.0 |
| BEIT-L-16-224 | IN21k+IN1k | self-supervised | 307.0 | 14.0 |
| BEITv2-B-16-224-1K-1K | IN1k+IN1k | self-supervised | 86.0 | 1.2 |
| BEITv2-B-16-224-22K-1K | IN1k+(IN21k+IN1k) | self-supervised | 86.0 | 14.0 |
| SIGLIP-SO400-14-384 | WebLI | contrastive | 878.0 | 22000.0 |
| SIGLIP2-SO400-14-384 | WebLI | contrastive | 1136.0 | 22000.0 |
| SIGLIP-B-16-224 | WebLI | contrastive | 203.0 | 22000.0 |
| SIGLIP2-B-16-224 | WebLI | contrastive | 375.0 | 22000.0 |

Table 3: Fine-tuning configurations for pre-trained models on ImageNet.

| Model | Epochs | LR | Batch Size | Weight Decay | Optimizer |
|---|---|---|---|---|---|
| ViT-MAE | 10 | $3 \times 10^{-4}$ | 32 | – | AdamW |
| DINOv1 | 5 | $1 \times 10^{-3}$ | 128 | – | AdamW |
| PaliGemma | 1 | $1 \times 10^{-3}$ | 64 | 0.01 | AdamW |

*[number of sides per polygon]-[number of polygons per row and column]-[number of concentric polygons] [color scheme].* This approach generates diverse geometric occlusion patterns that preserve semantic content while introducing systematic visual perturbations. Each mask variant was applied to the complete ImageNet validation set.

### B.4 Coco Objects

In the Coco Objects attack, we employed Facebook's DETR object detection model with a Resnet50 backbone to crop subject objects from the ImageNet validation split [6]. Each cropped result underwent manual review, with outcomes systematically recorded in CSV files organized by class for later reuse. Manual evaluation of 120 classes yielded 3378 images with correct cropping. Images

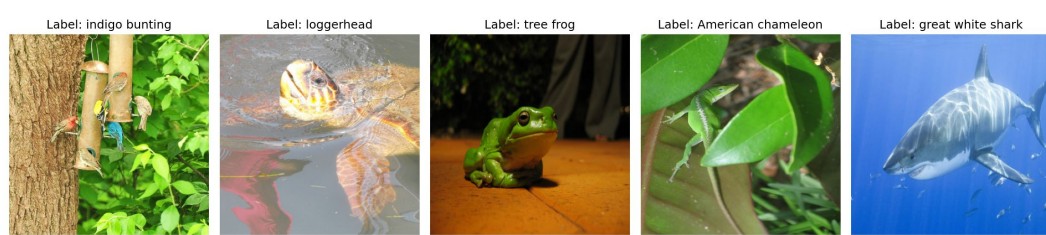

Figure 6: Sample images from the clean ImageNet1k validation split

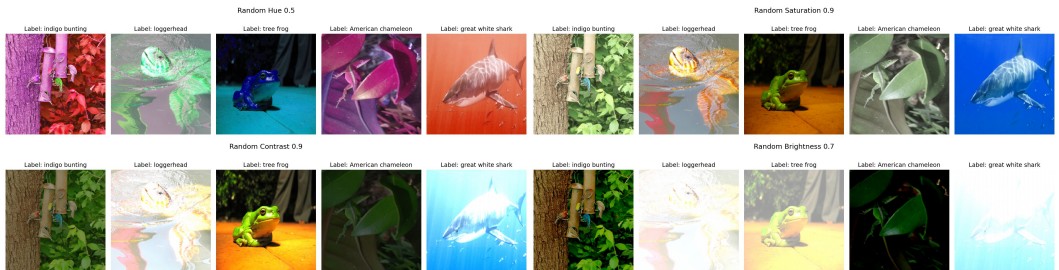

Figure 7: Sample images from the Random Perturbations attack

where the cropped mask occupied less than 1% or more than 60% of the original image area were subsequently filtered, resulting in a final dataset of 2055 images. The cropped subjects were then composited onto three background types: solid black, thresholded Perlin noise based on Yahya Jabary's implementation [29], and continuous Perlin noise, while preserving the original size and spatial positioning of the cropped elements. Attack success rates were computed using the unmodified original versions of the 2055 selected images as the baseline reference.

### B.5 ImageNet-C

ImageNet-C comprises 19 distinct perturbations categorized into noise, blur, weather, and digital corruption types [22]. Each corruption was systematically applied to the complete ImageNet validation split across five graduated severity levels, where severity 1 represents the lightest perturbation and severity 5 constitutes the most vigorous corruption intensity. The experimental protocol evaluated model performance at each severity level using randomly sampled subsets of 100,000 images drawn from across all 19 distinct perturbations to ensure computational feasibility while maintaining statistical validity.

### B.6 ImageNet-R

ImageNet-R consists of a 200-class subset derived from ImageNet, featuring a test set of 30,000 images that contain diverse artistic and stylistic renditions of standard object categories [23]. These renditions encompass various non-photographic representations, including paintings, embroidery, sketches, and other artistic interpretations, challenging model robustness to domain shift and stylistic variation while maintaining semantic content consistency with the original ImageNet classification task. As no direct clean counterparts exist for these artistic renditions, the attack success rate was approximated using ImageNet-200, the corresponding 200-class subset of the ImageNet validation dataset, as the clean baseline for measuring accuracy degradation [23]. Since ImageNet-R contains only 200 of the original 1000 ImageNet classes, model predictions for ImageNet-R and ImageNet-200 were restricted to these 200 classes.

## C  Sample Images of the Attacks Employed in the Robustness Analysis

The Random Perturbation attacks were applied as follows:

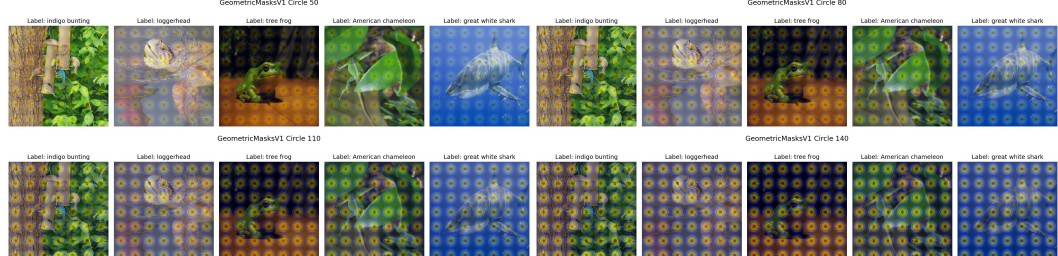

Figure 8: Sample images from the GeometricMasksV1 attack

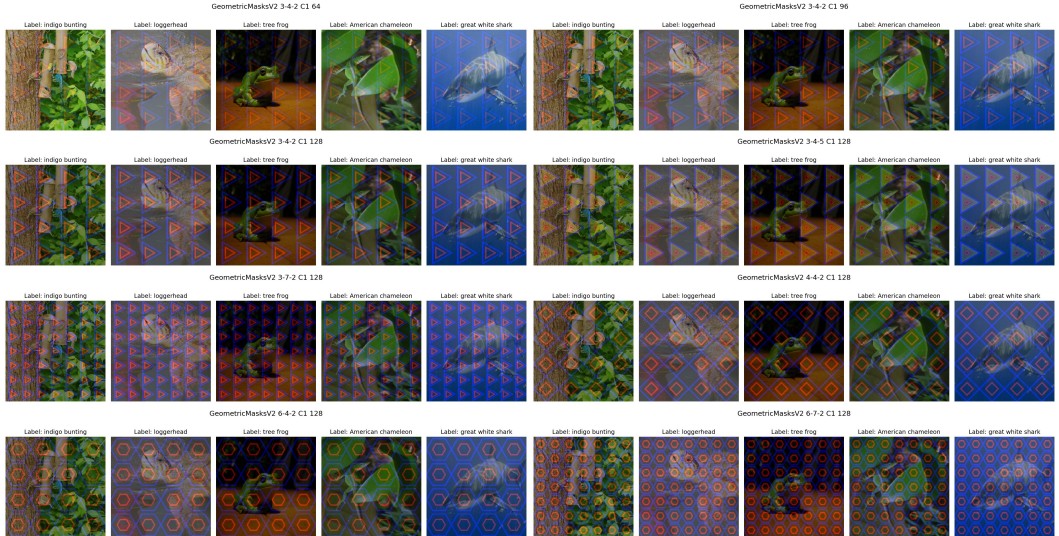

Figure 9: Sample images from the GeometricMasksV2 attack

- **Hue perturbation** ($\delta_h = 0.5$): Applies a random hue shift $h \sim \mathcal{U}(-0.5, 0.5)$ in HSV colour space, where values represent fractions of the full hue rotation cycle.

- **Saturation perturbation** ($\delta_s = 0.9$): Multiplies pixel saturation by a factor $s \sim \mathcal{U}(0.1, 1.9)$, enabling transitions from near-grayscale ($s = 0.1$) to highly saturated ($s = 1.9$) conditions.

- **Contrast perturbation** ($\delta_c = 0.9$): Adjusts image contrast through scaling pixel intensities $I' = I \times c$, where $c \sim \mathcal{U}(0.1, 1.9)$.

- **Brightness perturbation** ($\delta_b = 0.7$): Shifts pixel intensities $I' = I + b$ by a factor $b \sim \mathcal{U}(0.3, 1.7)$, uniformly modulating image luminance across all channels.

## D    Sample Images of the Adversarial Fine-Tuning Evaluation

The random mask in the fine-tuning of ResNet50-v3 was randomly chosen for each perturbed image from the configurations *[3, 4, 6, 10]-[2, 4, 7, 10]-[2, 5, 10] [C1, C2]* as shown in Figures 13 and 14.

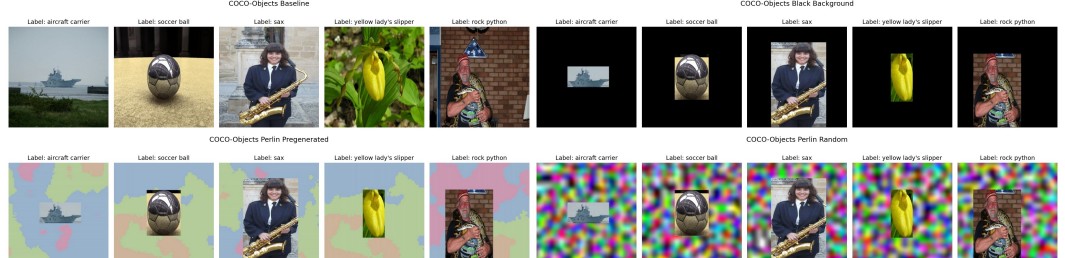

Figure 10: Sample images from the COCO Objects attack

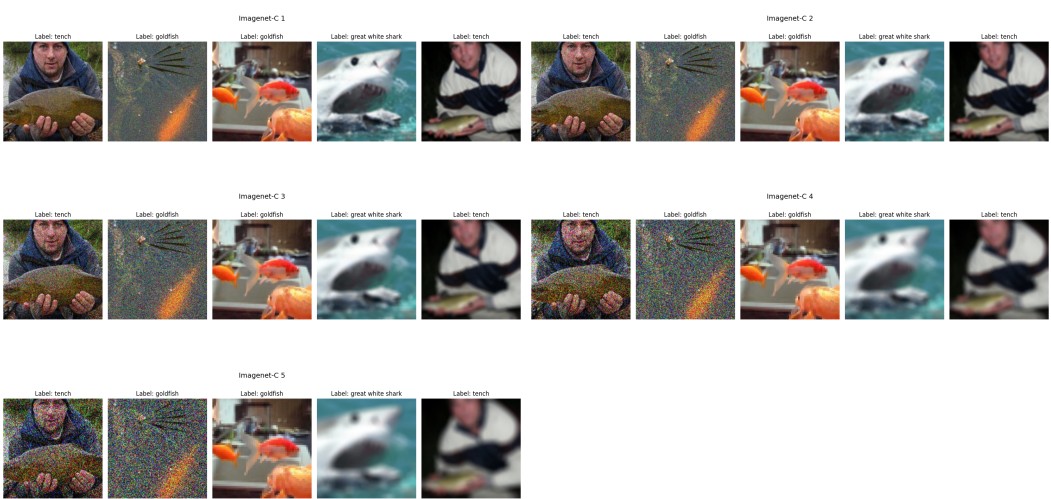

Figure 11: Sample images from the five severity levels in ImageNet-C

# E Human Evaluation GUI

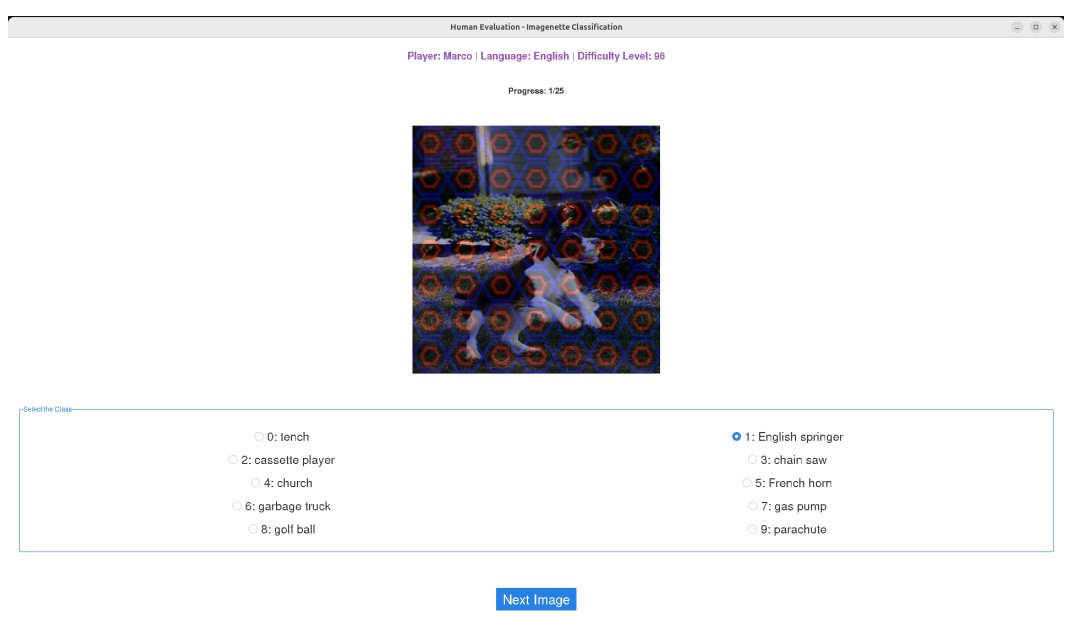

Figure 15: Graphical user interface (GUI) used for the human evaluation task

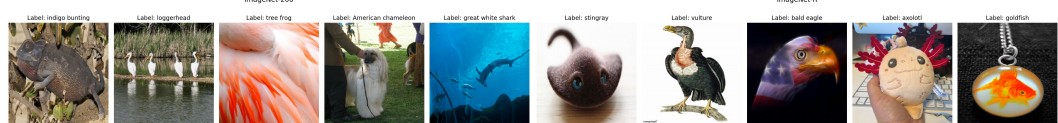

Figure 12: Sample images from ImageNet-200 and ImageNet-R

**Sample Images Adversarial Fine-Tuning Evaluation**

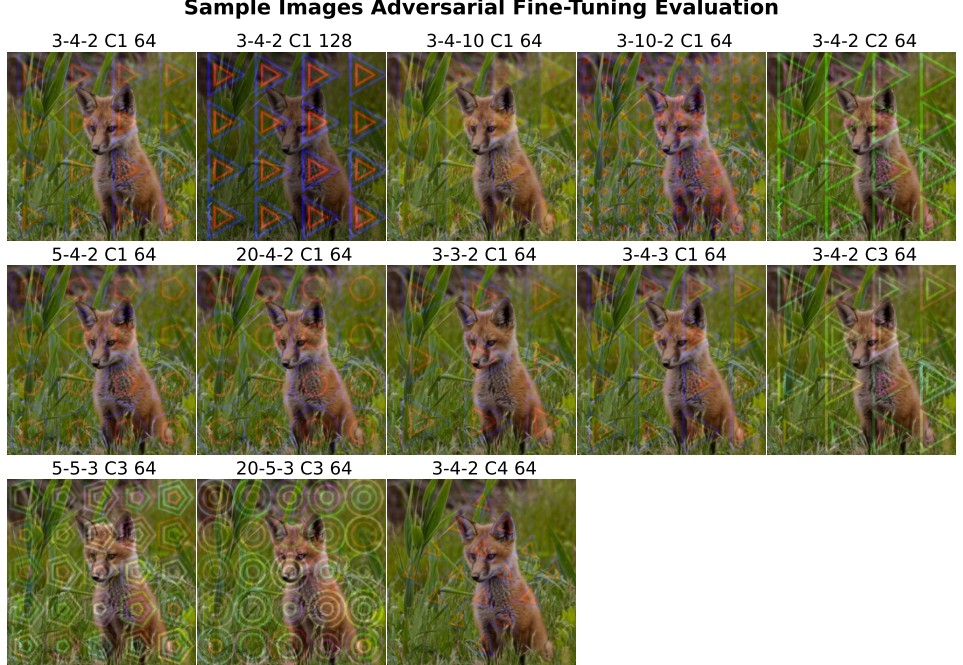

Figure 13: Sample images of the GeometricMasksV2 configurations used in the evaluation of the adversarially fine-tuned ResNet50s

# F Comparisons of Models

## F.1 Comparison of CLIP models

Table 4: Attack success rates (%) for selected CLIP-ViT-L-14 models

| Attack Category | Attack Type | VIT-L-14- METACLIP 400M | VIT-L-14 METACLIP FULLCC | APPLE DFN2B-CLIP-VIT-L-14 | EVA02-L-14 |
|---|---|---|---|---|---|
| Random | Hue 0.5 | 16.1 | 14.5 | 10.8 | 11.5 |
| | Saturation 0.9 | 4.1 | 3.5 | 2.2 | 2.3 |
| | Contrast 0.9 | 6.1 | 5.3 | 3.1 | 3.5 |
| | Brightness 0.7 | 15.0 | 13.8 | 9.7 | 8.7 |
| GeometricMasksV1 | Circle 50 | 23.2 | 18.8 | 15.0 | 11.6 |
| | Circle 80 | 43.2 | 35.3 | 35.5 | 24.9 |
| | Circle 110 | 63.8 | 55.6 | 63.2 | 44.9 |
| | Circle 140 | 83.0 | 77.3 | 87.7 | 69.8 |
| GeometricMasksV2 | 3-4-2 C1 Opacity 64 | 17.2 | 14.1 | 11.2 | 10.1 |
| | 3-4-2 C1 Opacity 96 | 22.7 | 19.6 | 18.4 | 15.2 |
| | 3-4-2 C1 Opacity 128 | 28.8 | 25.8 | 27.3 | 21.8 |
| | 3-4-5 C1 Opacity 128 | 30.8 | 26.8 | 28.4 | 22.3 |
| | 3-7-2 C1 Opacity 128 | 43.1 | 37.0 | 47.5 | 34.2 |
| | 6-4-2 C1 Opacity 128 | 48.8 | 51.3 | 53.2 | 42.4 |
| | 6-7-2 C1 Opacity 128 | 69.0 | 72.7 | 79.0 | 62.2 |
| COCO Objects | Black Background | 20.5 | 19.0 | 16.4 | 13.6 |
| | Thresholded Perlin Noise Background | 22.0 | 21.0 | 17.9 | 15.8 |
| | Perlin Noise Background | 27.5 | 24.0 | 21.4 | 17.2 |
| ImageNet-R | ImageNet-R | 9.5 | 8.3 | 9.0 | 6.3 |
| ImageNet-C | Distortion Severity 1 | 10.4 | 8.5 | 6.7 | 5.4 |
| | Distortion Severity 2 | 18.7 | 16.1 | 13.5 | 9.4 |
| | Distortion Severity 3 | 25.8 | 22.0 | 19.3 | 13.2 |
| | Distortion Severity 4 | 37.9 | 32.4 | 28.7 | 19.4 |
| | Distortion Severity 5 | 51.6 | 45.5 | 41.6 | 29.6 |
| Overall Average | Mean of Averages per Attack Category | 27.1 | 24.3 | 24.0 | 18.5 |

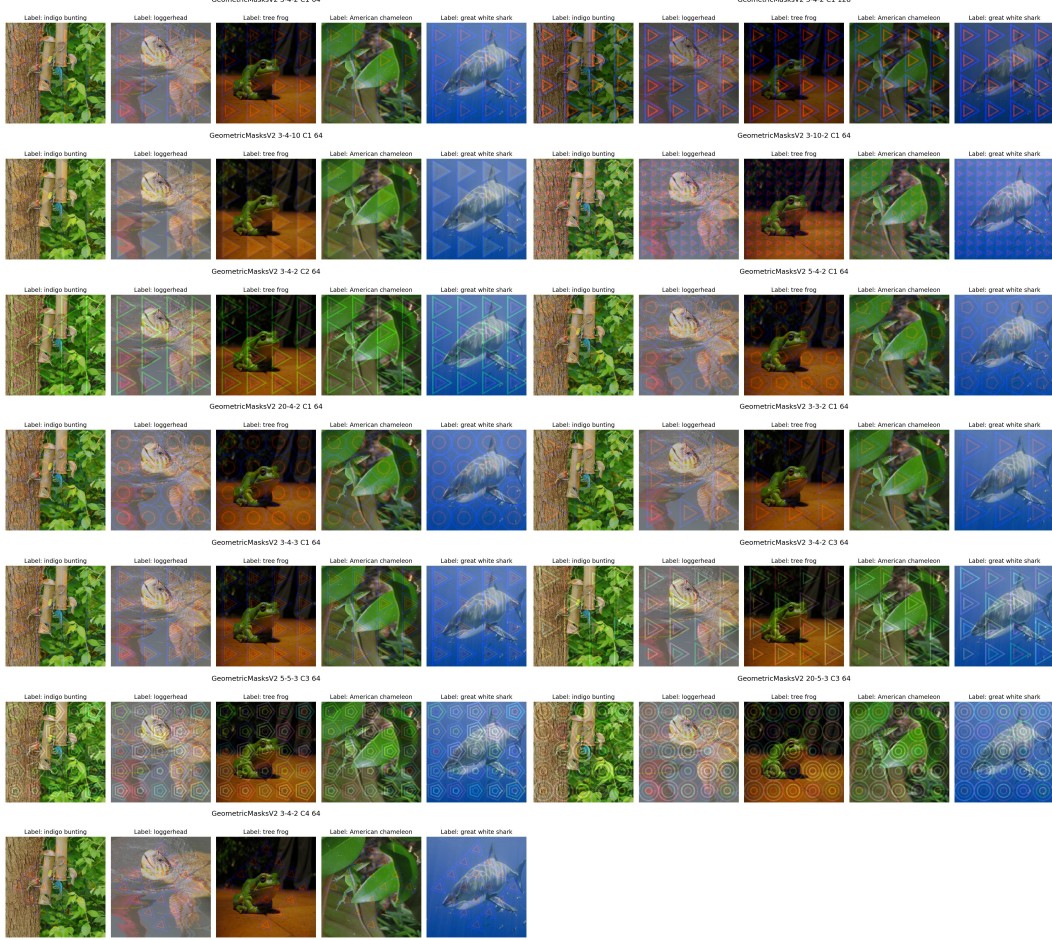

Figure 14: Sample images from the GeometricMasksV2 attack used for the ResNet50 fine-tuning and validation

Four CLIP-ViT-L-14 variants with identical architectures but distinct training datasets reveal attack-specific vulnerability patterns that illuminate the relationship between data curation strategies and robustness mechanisms in Table 4.

Random Perturbations expose fundamental differences in color invariance learning. EVA02-L-14 and DFN-2B demonstrate superior resilience, while MetaClip variants show almost doubled vulnerability. The stark contrast in hue and brightness perturbation resistance suggests that multi-source curation (LAION-2B + COYO-700M) and quality filtering (DFN-2B) better preserve color and brightness consistency during training than metadata-based selection alone.

In the GeometricMasksV1 category, EVA02-L-14 maintains a significant advantage, and MetaCLIP FullCC consistently outperforms the 400M version. DFN-2B unexpectedly deteriorates increasingly, starting at occlusion 80, exceeding even MetaClip-400M. This vulnerability inversion indicates that DFN-2B's quality filtering may systematically exclude partially occluded objects, creating blind spots that manifest under severe geometric perturbations.

GeometricMasksV2 attacks demonstrate complex pattern-dependent vulnerabilities. EVA02-L-14 maintains the lowest attack success rates with an advantage of more than 10%. DFN-2B again shows deteriorated performance on opacities above 96. The MetaCLIP models perform similarly to DFN-2B, but surprisingly, the masks 6-4-2 C1 and 6-7-2 C2 invert the ordering of the MetaCLIP models' performance, with the 400M version outperforming the FullCC one.

COCO Objects manipulations show relatively compressed performance differences (13.6-27% ASR range). EVA02-L-14 performs the best, with DFN-2B having an ASR increased by about 2-3%. The

MetaCLIP models show an even higher vulnerability. While the MetaCLIP FullCC outperforms the 400M version, it has an ASR which is between 32% and 39% higher than that of EVA02-L-14.

ImageNet-R produces the tightest clustering of results (6.3-9.5% ASR), suggesting that artistic domain shifts probe fundamental visual representations largely independent of training data characteristics. However, one has to keep in mind that all four models have been trained on relatively large datasets. The minimal variation indicates that current CLIP training strategies, regardless of scale or curation method, develop similar capabilities for handling stylistic variations.

ImageNet-C corruptions reveal progressive differentiation with severity. At low severities, performance differences remain modest (at most 5% at severity 1), but diverge substantially at maximum corruption (29.6-51.6% at severity 5, a relative increase of up to 74%). EVA02-L-14's consistent advantage across all severity levels—maintaining sub-30% ASR even at severity 5—demonstrates that combining high-quality datasets from multiple sources builds more robust representations against naturalistic corruptions than single-source approaches.

These attack-specific patterns establish that training data characteristics influence robustness mechanisms differentially across perturbation types. Multi-source curation (EVA02-L-14) provides consistent advantages across all attack categories, while quality-focused filtering (DFN-2B) creates asymmetric robustness profiles with specific vulnerabilities to geometric occlusions. Scale without quality control (MetaClip Full CC) offers minimal benefits, indicating that strategic data curation supersedes volume for comprehensive adversarial robustness.

## F.2 Comparison of BEiTv2 Models

Table 5: Attack success rates (%) for selected BEiTv2 models

| Attack Category | Attack Type | BEITV2 B 16 224 IN1K FT IN1K | BEITV2 B 16 224 IN1K FT IN22K IN1K |
|---|---|---|---|
| Random | Hue 0.5 | 7.7 | 7.9 |
| | Saturation 0.9 | 1.2 | 1.1 |
| | Contrast 0.9 | 1.5 | 1.3 |
| | Brightness 0.7 | 5.8 | 5.5 |
| GeometricMasksV1 | Circle 50 | 11.0 | 7.0 |
| | Circle 80 | 19.7 | 11.5 |
| | Circle 110 | 34.5 | 24.5 |
| | Circle 140 | 58.4 | 54.4 |
| GeometricMasksV2 | 3-4-2 C1 Opacity 64 | 8.8 | 5.7 |
| | 3-4-2 C1 Opacity 96 | 12.9 | 9.4 |
| | 3-4-2 C1 Opacity 128 | 17.9 | 20.7 |
| | 3-4-5 C1 Opacity 128 | 19.6 | 18.5 |
| | 3-7-2 C1 Opacity 128 | 32.4 | 29.8 |
| | 6-4-2 C1 Opacity 128 | 77.9 | 49.9 |
| | 6-7-2 C1 Opacity 128 | 68.7 | 69.0 |
| COCO Objects | Black Background | 9.8 | 10.8 |
| | Thresholded Perlin Noise Background | 15.1 | 11.8 |
| | Perlin Noise Background | 61.1 | 23.9 |
| ImageNet-R | ImageNet-R | 35.2 | 30.8 |
| ImageNet-C | Distortion Severity 1 | 7.7 | 6.6 |
| | Distortion Severity 2 | 12.1 | 11.2 |
| | Distortion Severity 3 | 17.4 | 14.5 |
| | Distortion Severity 4 | 24.6 | 20.7 |
| | Distortion Severity 5 | 36.2 | 31.1 |
| Overall Average | Mean of Averages per Attack Category | 25.4 | 20.1 |

The comparison between BEiTv2 Base Patch-16 models fine-tuned exclusively on ImageNet-1K versus sequential fine-tuning on ImageNet-21K followed by ImageNet-1K in Table 5 reveals consistent improvements in adversarial robustness. The extended fine-tuning protocol reduces the overall average ASR from 25.4% to 20.1%, representing a relative improvement of 20.8%.

Random color perturbations show low ASRs for both models, with the models performing almost identically, indicating that the small ImageNet-1K suffices against these low-level vulnerabilities, and expanding the dataset with ImageNet-21K does not lead to significant improvements.

Extended fine-tuning demonstrates the most substantial benefits against geometric occlusions. For GeometricMasksV1, the improvement scales with perturbation severity: minimal gains at low opacity expand up to higher occlusions of 110. The most striking improvement occurs in GeometricMasksV2 attacks, particularly for the 6-4-2 C1 configuration, where ASR decreases from 77.9% to 49.9%.

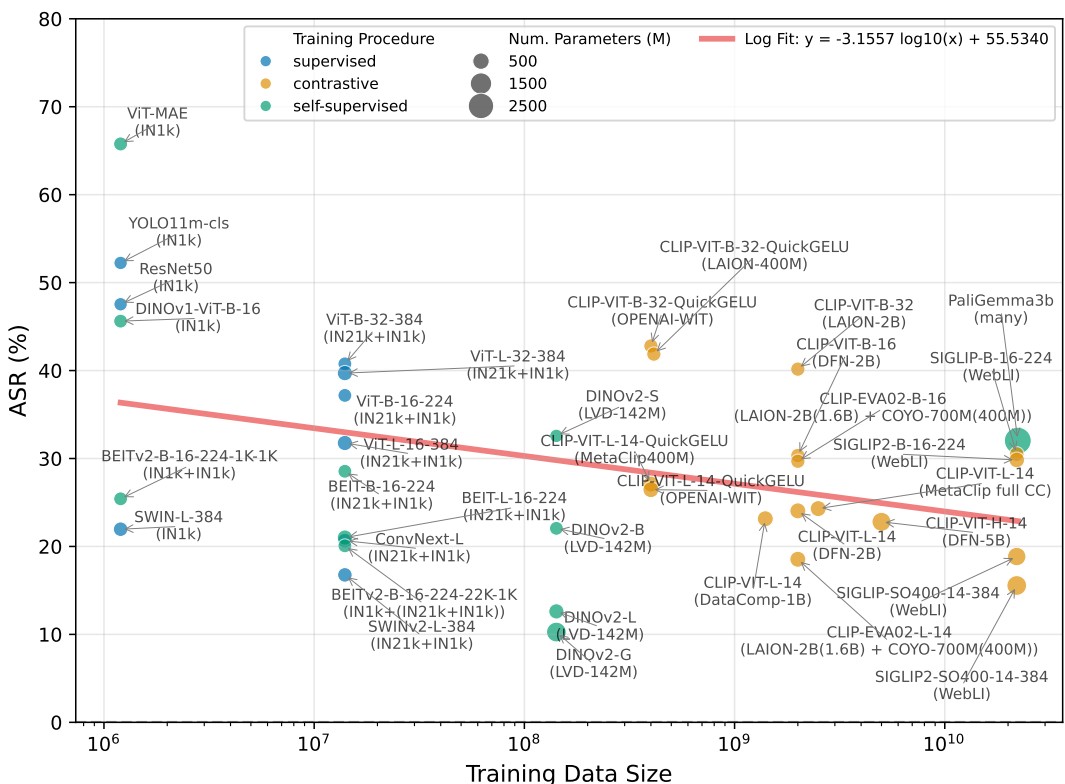

Figure 16: Overall average ASR relative to the size of the training data averaged across: Random Perturbation, GeometricMasksV1, GeometricMasksV2, Coco Objects, ImageNet-C, and ImageNet-R attacks.

However, for the other configurations in the GeometricMasksV2 category, there is little to no improvement.

Background manipulation attacks reveal selective improvements. While performance on black and thresholded Perlin backgrounds shows modest gains, the most significant reduction occurs with continuous Perlin noise backgrounds, indicating that extended fine-tuning particularly strengthens robustness to complex textural perturbations. This asymmetric improvement pattern suggests that ImageNet-21K exposure specifically addresses vulnerabilities to high-frequency noise patterns.

The benefits extend uniformly across corruption severities in ImageNet-C, with consistent relative improvements of 13-16% across all levels. Similarly, ImageNet-R shows a reduction from 35.23% to 30.78%, demonstrating that the expanded fine-tuning enhances generalization to artistic domain shifts.

These results indicate that hierarchical fine-tuning on progressively focused datasets (ImageNet-21K and ImageNet-1K) provides a more robust feature hierarchy than direct ImageNet-1K fine-tuning, particularly for spatially structured perturbations and complex backgrounds, while maintaining competitive performance on standard corruptions.

# G   Extra/detailed results

## G.1   Overall results with labels for all points

## G.2   Attack success rate per attack type

The attack success rates exhibit markedly different patterns across attack types, with the fitted logarithmic curves revealing distinct vulnerabilities in model robustness. While Random Perturbations and ImageNet-C demonstrate moderate declining trends in Figures 18a and 18e, ImageNet-R in

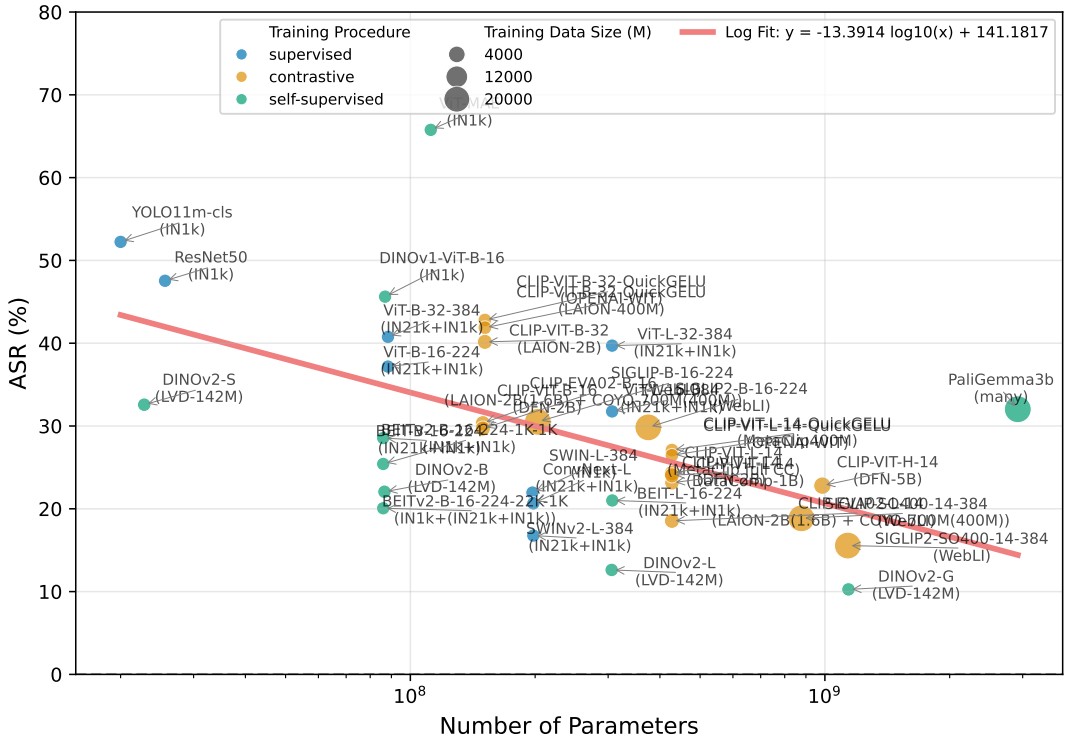

Figure 17: Overall average ASR relative to the number of model parameters averaged across: Random Perturbation, GeometricMasksV1, GeometricMasksV2, Coco Objects, ImageNet-C, and ImageNet-R attacks.

Figure 18f shows a dramatically steeper decrease, indicating that increased training data provides substantially greater protection against stylistic domain shifts than against common corruptions. This divergence likely stems from the fundamental nature of these attacks: ImageNet-C introduces perturbations such as noise, blur, and weather effects that are typically absent from standard training datasets, resulting in consistent vulnerability across models regardless of training data scale. In contrast, ImageNet-R's artistic renditions and alternative representations may be partially captured within large-scale training corpora, enabling models trained on extensive datasets to develop more robust features for handling stylistic variations. The intermediate trends observed in geometric mask attacks and COCO Objects perturbations suggest that structured occlusions and background manipulations represent a middle ground, where training data scale provides meaningful but limited improvements in robustness.

### G.2.1 Training Procedure and Adversarial Robustness

Looking at Figure 19, showing the overall average attack success rates by training procedure, the relationship between training methodology and adversarial robustness appears surprisingly limited. Contrastive learning shows the lowest average ASR at 27.9%, indicating only a modest benefit from cross-modal alignment in learning robust representations. Self-supervised learning shows similar performance to contrastive learning at 28.4%, while supervised learning shows the highest vulnerability with 34.3% on average. However, the wide standard deviations reveal significant overlap across all three training paradigms, which suggests that implementation details and architectural choices may have a greater impact than the core training procedure itself.

### G.3 Dataset Source and Adversarial Robustness

Figure 20 compares the scaling behavior of ASR between models trained on web-crawled and non-web-crawled datasets. For non-web-crawled datasets, the fitted relationship is ASR $= -4.3797 \log_{10}(x) + 63.1610$ while for web-crawled datasets it is ASR $= -4.4031 \log_{10}(x) + 69.9639$.

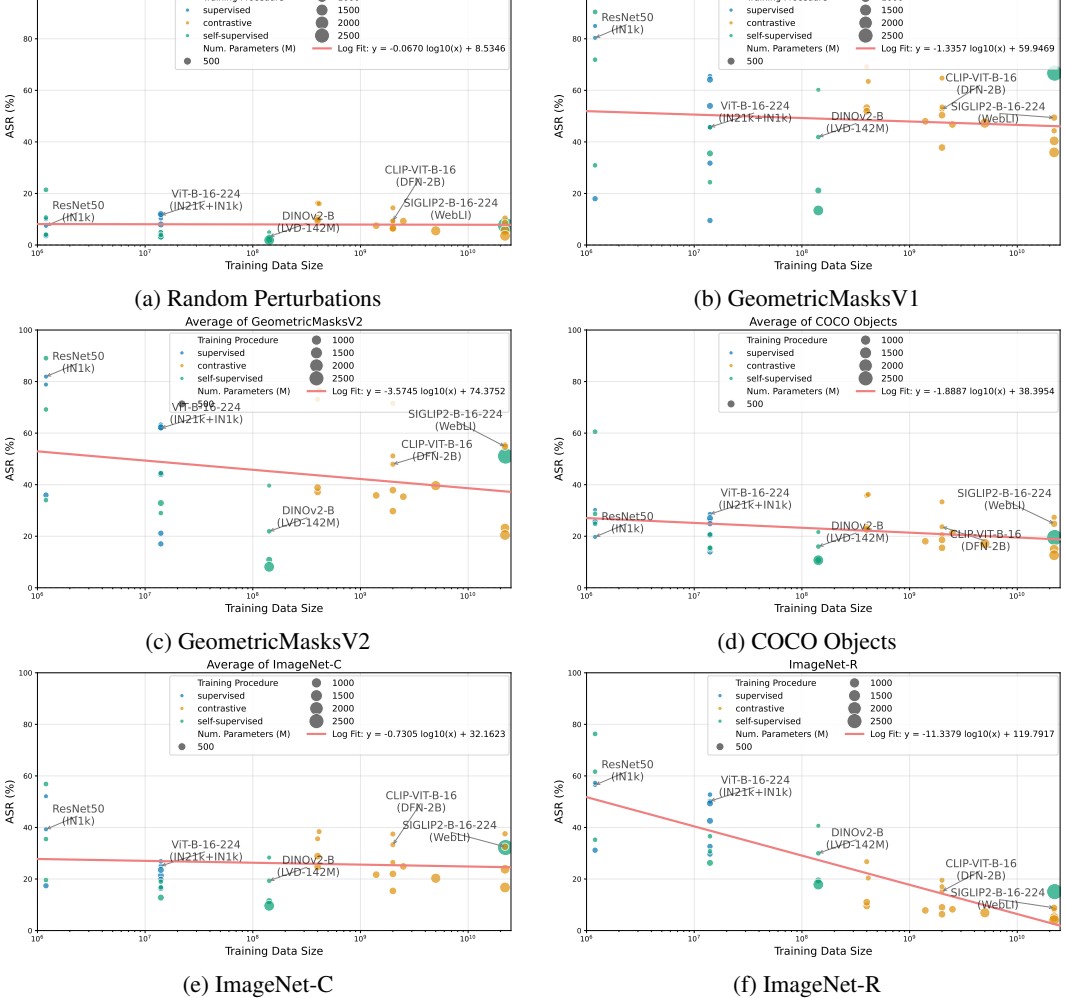

Figure 18: Average attack success rates per attack type

At first sight, the two fitted functions appear nearly identical, differing only slightly in slope and intercept. According to this estimate, models trained on non-web-crawled datasets would require in the order of $2.63 \times 10^{14}$ examples ($\approx 263$ trillion) to achieve an ASR of zero. In contrast, models trained on web-crawled data would require roughly $7.65 \times 10^{15}$ examples ($\approx 7.65$ quadrillion). This corresponds to a factor of about $29\times$ more training data for web-crawled models to reach the same level of robustness. These results make clear that raw scale alone cannot close the robustness gap.

## G.4 Worst-Case Analysis of Geometric masks

Figure 21 shows the maximum attack success rates (ASR) for the two GeometricMasks categories across a wide set of models. The maximum attack success rate of our robustness scaling analysis resulted from either the category GeometricMasksV1 or GeometricMasksV2 for all evaluated models. For GeometricMasksV1, the Circle 140 variant consistently achieved the highest ASR across all models, whereas in GeometricMasksV2, the worst case was almost always the 6-7-2 C1 Opacity 128 variant, with 6-4-2 C1 Opacity 128 occasionally dominating. Notably, the same attack variant tended to yield the worst case of our attacks within entire model families, e.g., CLIP, DINOv2, SWIN, BEiT, BEiTv2, indicating that adversarial vulnerabilities are not randomly distributed but reflect systematic weaknesses tied to architectural or training similarities. From the perspective of an adversary, this means that even without precise knowledge of the deployed model, limited information about the model family can already guide the choice of attack: knowing the family suffices to select a

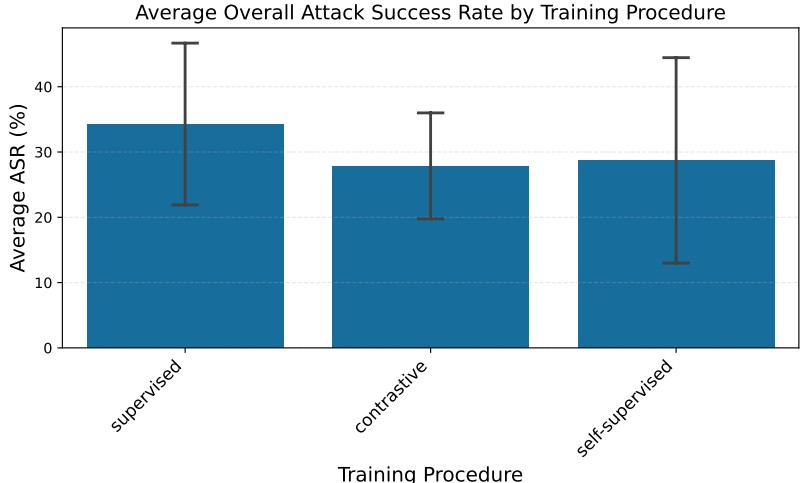

Figure 19: Average attack success rates by training procedure with standard deviation error bars

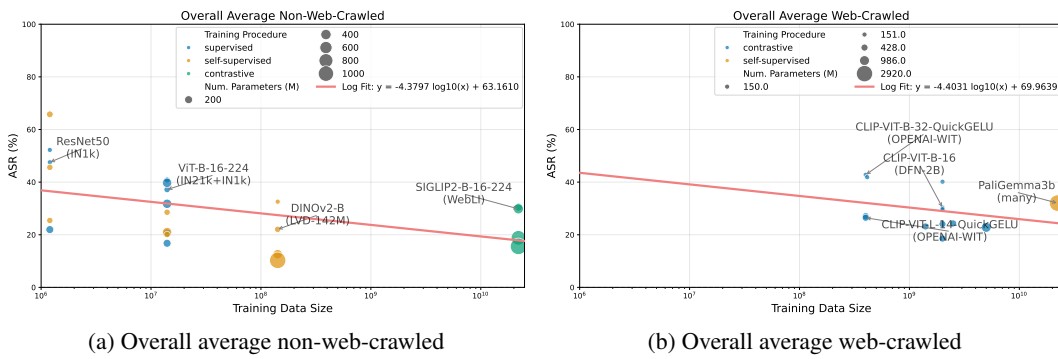

(a) Overall average non-web-crawled

(b) Overall average web-crawled

Figure 20: Robustness comparison between non-web-crawled and web-crawled datasets

variant that is likely to perform near-optimally across all its members. This family-level consistency substantially reduces the uncertainty an attacker would face in practice and highlights the need to consider family-specific robustness evaluations. Furthermore, these findings show that DINOv2 not only achieves strong robustness on average but also maintains resilience under worst-case adversarial scenarios, underscoring its relative reliability across both typical and extreme conditions.

## G.5 Adversarial Fine-Tuning

The evaluation of fine-tuned ResNet50 models, alongside the vanilla ResNet50, reveals distinct patterns in how adversarial training with geometric masks influences model robustness and generalization capabilities. Figure 22 presents the Attack Success Rates (ASR) across various mask configurations, demonstrating the effectiveness of different training strategies. The accuracies of the fine-tuned models on the clean ImageNet validation dataset lie between 75% and 77%.

### G.5.1 Generalization Across Opacity Levels

Fine-tuning with geometric masks demonstrates remarkable generalization across opacity levels for all models. The Model ResNet50-v1, trained on opacity 64, exhibits only slightly decreased performance when evaluated on opacity 128. In contrast, the vanilla ResNet50 displays an extreme vulnerability to opacity variations. This difference underscores the effectiveness of adversarial fine-tuning in creating opacity-invariant representations.

### G.5.2 Structural Generalization

The fine-tuned models demonstrate robust generalization across geometric variations:

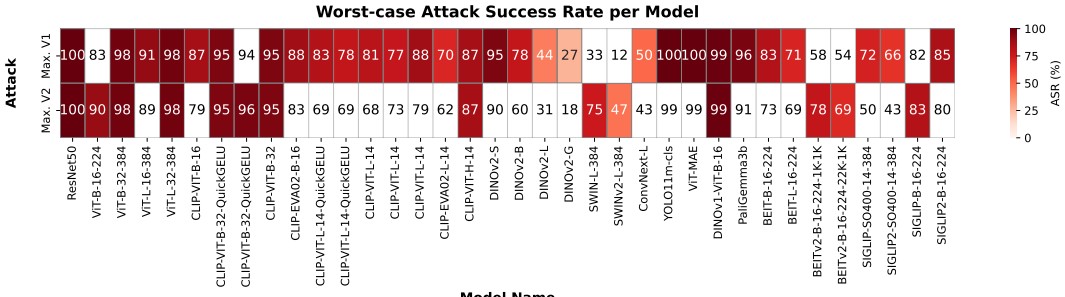

Figure 21: Worst-case attack success rate of the robustness analysis. "Max. V1" is the maximum ASR that occurred in the GeometricMasksV1 attack for a given model, where only the maximum is colored. "Max. V2" signifies GeometricMasksV2

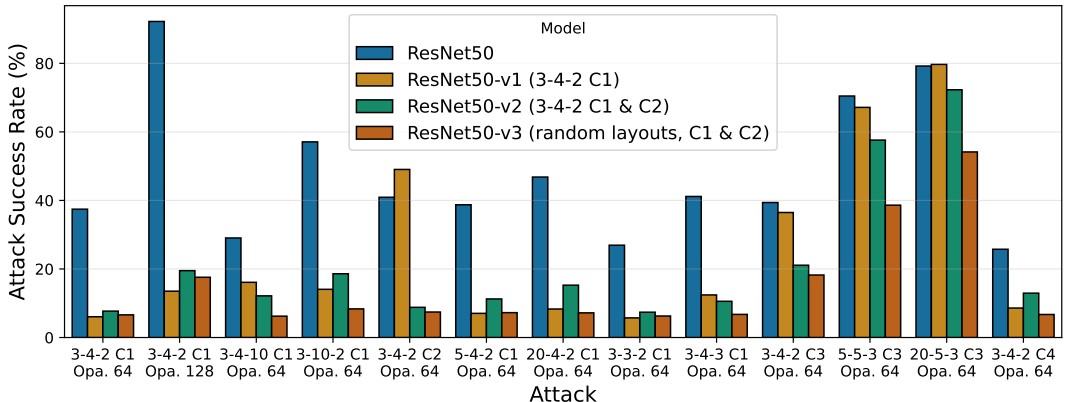

Figure 22: Attack success rate comparison between the vanilla ResNet50 and three fine-tuned variants on various GeometricMasksV2 attacks. The attacks are indicated by their masks (a–b–c), color scheme (C1, C2, C3, or C4), and the opacity of the masks.

**Polygon sides** Models maintain consistent performance across masks with varying polygon sides. The ASR remains low for both 5-sided (5-4-2 C1) and 20-sided (20-5-3 C1) polygon configurations, with all fine-tuned variants achieving ASRs below 16%, while the vanilla ResNet50 shows ASRs of 38.7% and 46.8%, respectively.

**Polygons per row and column** Varying the number of polygons per row and column has minimal impact on fine-tuned model performance. Masks 3-10-2 C1 (high column density) and 3-3-2 C1 (low density) yield similar ASRs across all fine-tuned models. In contrast, the high column density configuration resulted in the fourth highest ASR for the standard ResNet50.

**Concentric polygons** The models effectively generalize across different numbers of concentric layers, as evidenced by consistent performance on masks 3-4-10 C1 and 3-4-3 C1. Notably, on the 3-4-3 C1 mask, the attack success rate of the vanilla ResNet50 was 3.3 times higher than that of the worst-performing fine-tuned model, highlighting the robustness gains achieved through fine-tuning.

**Rotation** The 3-4-2 C4 configuration, rotated 45 degrees from the standard orientation, shows minimal performance degradation across fine-tuned models, indicating learned rotation-invariant features. However, it is essential to note that this was also the weakest attack against the vanilla ResNet50, with an ASR of only 25.8%, presumably because the mask does not cover the entire image.

### G.5.3 Color Scheme Limitations

Color generalization represents the primary limitation of the fine-tuning approach. Models exhibit strong performance primarily on color schemes encountered during training:

Table 6: Accuracy (%) of models and humans at different opacity levels. These are the values shown in Figure 5.

| Opacity level | 0 | 64 | 96 | 128 |
|---|---|---|---|---|
| ResNet50 | 99.90 | 89.32 | 52.05 | 26.17 |
| ResNet50-v1 (3-4-2 C1) | 99.85 | 99.13 | **97.43** | 87.16 |
| ViT-B-16-224 | 99.72 | 98.24 | 90.04 | 59.49 |
| CLIP-VIT-B-16 | 99.34 | 96.31 | 87.36 | 69.12 |
| DINOv2-B | 99.97 | **99.26** | 95.82 | 86.52 |
| Avg. Humans | **100.00** | 97.33 | 96.00 | **93.33** |

ResNet50-v1, trained exclusively on color scheme C1, shows elevated ASRs when evaluated on C2 and C3 color schemes. ResNet50-v2, trained on both C1 and C2 color schemes, demonstrates improved generalization with lower ASRs across both schemes. On the novel C3 color scheme (unseen during training), all fine-tuned models show degraded performance. However, the models v2 and v3 are less affected than v1.

ResNet50-v4, trained with random mask selection and dual color schemes, achieves the most consistent performance across all evaluations. On the configurations 5-5-3 C3 and 20-5-3 C3, which have all parameters different from those seen during training, ResNet50-v4 outperforms the other models by a significant margin of 18%, suggesting that diverse training conditions promote broader generalization.

These results indicate that while geometric and structural features can be effectively learned through adversarial fine-tuning, color-based robustness requires explicit exposure to diverse color schemes during training, highlighting the importance of comprehensive augmentation strategies.

### G.6 Table with human and model performance

## H Full Adversarial Robustness Evaluation Results

## I Compute usage

The experiments were conducted on an internal cluster equipped with RTX 3090s and RTX 2080 TIs. In total, we have logged 322.5 GPU hours for the experiments and testing.


Table 7: Baseline accuracies (%) for ImageNet1K-Val, ImageNet-200, and Images in COCO Objects

| Model Name | Training Dataset | ImageNet1K-Val | ImageNet-200 | Images in COCO Objects |
|---|---|---|---|---|
| ResNet50 | IN1k | 80.0 | 93.8 | 78.9 |
| ViT-B-16-224 | IN21k+IN1k | 80.3 | 94.2 | 79.1 |
| ViT-B-32-384 | IN21k+IN1k | 81.2 | 94.8 | 80.9 |
| ViT-L-16-384 | IN21k+IN1k | 85.0 | 96.3 | 83.8 |
| ViT-L-32-384 | IN21k+IN1k | 81.0 | 95.0 | 79.2 |
| CLIP-VIT-B-16 | DFN-2B | 74.0 | 93.4 | 71.3 |
| CLIP-VIT-B-32-QuickGELU | OPENAI-WIT | 59.6 | 85.8 | 60.2 |
| CLIP-VIT-B-32-QuickGELU | LAION-400M | 59.4 | 84.7 | 55.1 |
| CLIP-VIT-B-32 | LAION-2B | 63.7 | 87.6 | 62.4 |
| CLIP-EVA02-B-16 | LAION-2B(1.6B) + COYO-700M(400M) | 72.6 | 93.3 | 71.2 |
| CLIP-VIT-L-14-QuickGELU | MetaClip400M | 72.3 | 92.3 | 70.4 |
| CLIP-VIT-L-14-QuickGELU | OPENAI-WIT | 70.8 | 92.7 | 69.8 |
| CLIP-VIT-L-14 | DataComp-1B | 76.9 | 94.5 | 74.5 |
| CLIP-VIT-L-14 | MetaClip full CC | 74.2 | 94.4 | 72.4 |
| CLIP-VIT-L-14 | DFN-2B | 78.9 | 95.5 | 76.7 |
| CLIP-EVA02-L-14 | LAION-2B(1.6B) + COYO-700M(400M) | 77.4 | 95.6 | 75.0 |
| CLIP-VIT-H-14 | DFN-5B | 81.3 | 96.2 | 76.8 |
| DINOv2-S | LVD-142M | 80.9 | 94.5 | 79.6 |
| DINOv2-B | LVD-142M | 84.4 | 96.2 | 82.5 |
| DINOv2-L | LVD-142M | 86.2 | 97.2 | 84.5 |
| DINOv2-G | LVD-142M | 86.7 | 97.3 | 85.0 |
| SWIN-L-384 | IN1k | 86.6 | 97.3 | 85.1 |
| SWINv2-L-384 | IN21k+IN1k | 87.2 | 97.4 | 84.4 |
| ConvNext-L | IN21k+IN1k | 85.7 | 97.1 | 84.8 |
| YOLO11m-cls | IN1k | 77.3 | 92.8 | 76.8 |
| ViT-MAE | IN1k | 52.3 | 72.5 | 52.2 |
| DINOv1-ViT-B-16 | IN1k | 75.9 | 91.9 | 75.0 |
| PaliGemma3b | many | 83.7 | 95.8 | 83.4 |
| BEIT-B-16-224 | IN21k+IN1k | 84.5 | 96.4 | 83.5 |
| BEIT-L-16-224 | IN21k+IN1k | 87.2 | 97.4 | 84.5 |
| BEITv2-B-16-224-1K-1K | IN1k+IN1k | 85.5 | 96.8 | 84.6 |
| BEITv2-B-16-224-22K-1K | IN1k+(IN21k+IN1k) | 86.2 | 97.1 | 84.7 |
| SIGLIP-SO400-14-384 | WebLI | 80.8 | 96.3 | 79.9 |
| SIGLIP2-SO400-14-384 | WebLI | 74.1 | 90.0 | 76.8 |
| SIGLIP-B-16-224 | WebLI | 73.2 | 92.8 | 70.6 |
| SIGLIP2-B-16-224 | WebLI | 69.2 | 87.9 | 70.6 |

Question: Does the paper discuss the limitations of the work performed by the authors?

Answer: [Yes]

Justification: As part of the conclusion.

Guidelines:

- The answer NA means that the paper has no limitation while the answer No means that the paper has limitations, but those are not discussed in the paper.
- The authors are encouraged to create a separate "Limitations" section in their paper.
- The paper should point out any strong assumptions and how robust the results are to violations of these assumptions (e.g., independence assumptions, noiseless settings, model well-specification, asymptotic approximations only holding locally). The authors should reflect on how these assumptions might be violated in practice and what the implications would be.
- The authors should reflect on the scope of the claims made, e.g., if the approach is only tested on a few datasets or with a few runs. In general, empirical results often depend on implicit assumptions, which should be articulated.
- The authors should reflect on the factors that influence the performance of the approach. For example, a facial recognition algorithm may perform poorly when image resolution is low or images are taken in low lighting. Or a speech-to-text system might not be used reliably to provide closed captions for online lectures because it fails to handle technical jargon.

Table 8: Attack success rates (%) for Random Attacks

| Model Name | Training Dataset | Random Hue 0.5 | Random Saturation 0.9 | Random Contrast 0.9 | Random Brightness 0.7 | Avg. Random Perturbations |
|---|---|---|---|---|---|---|
| ResNet50 | IN1k | 14.6 | 2.3 | 2.8 | 10.0 | 7.4 |
| ViT-B-16-224 | IN21k+IN1k | 18.8 | 4.2 | 6.6 | 16.7 | 11.6 |
| ViT-B-32-384 | IN21k+IN1k | 16.8 | 3.7 | 5.5 | 15.4 | 10.3 |
| ViT-L-16-384 | IN21k+IN1k | 12.8 | 2.8 | 4.6 | 11.8 | 8.0 |
| ViT-L-32-384 | IN21k+IN1k | 19.4 | 4.7 | 7.4 | 16.5 | 12.0 |
| CLIP-VIT-B-16 | DFN-2B | 16.0 | 3.6 | 4.3 | 13.8 | 9.4 |
| CLIP-VIT-B-32-QuickGELU | OPENAI-WIT | 27.5 | 7.6 | 9.4 | 20.8 | 16.3 |
| CLIP-VIT-B-32-QuickGELU | LAION-400M | 25.8 | 7.1 | 9.4 | 21.8 | 16.0 |
| CLIP-VIT-B-32 | LAION-2B | 23.9 | 6.6 | 7.7 | 19.5 | 14.4 |
| CLIP-EVA02-B-16 | LAION-2B(1.6B) + COYO-700M(400M) | 16.2 | 3.6 | 4.6 | 12.8 | 9.3 |
| CLIP-VIT-L-14-QuickGELU | MetaClip400M | 16.1 | 4.1 | 6.1 | 15.0 | 10.3 |
| CLIP-VIT-L-14-QuickGELU | OPENAI-WIT | 17.0 | 3.6 | 5.1 | 12.6 | 9.6 |
| CLIP-VIT-L-14 | DataComp-1B | 13.4 | 2.7 | 3.3 | 10.5 | 7.5 |
| CLIP-VIT-L-14 | MetaClip full CC | 14.5 | 3.5 | 5.3 | 13.8 | 9.3 |
| CLIP-VIT-L-14 | DFN-2B | 10.8 | 2.2 | 3.1 | 9.7 | 6.5 |
| CLIP-EVA02-L-14 | LAION-2B(1.6B) + COYO-700M(400M) | 11.5 | 2.3 | 3.5 | 8.7 | 6.5 |
| CLIP-VIT-H-14 | DFN-5B | 8.7 | 1.9 | 2.7 | 8.7 | 5.5 |
| DINOv2-S | LVD-142M | 7.0 | 1.9 | 2.1 | 8.7 | 4.9 |
| DINOv2-B | LVD-142M | 4.3 | 1.3 | 1.3 | 6.1 | 3.2 |
| DINOv2-L | LVD-142M | 2.5 | 0.9 | 0.8 | 4.0 | 2.0 |
| DINOv2-G | LVD-142M | 2.1 | 0.8 | 0.8 | 3.8 | 1.9 |
| SWIN-L-384 | IN1k | 7.2 | 1.2 | 1.4 | 4.7 | 3.6 |
| SWINv2-L-384 | IN21k+IN1k | 6.7 | 0.9 | 1.1 | 3.9 | 3.2 |
| ConvNext-L | IN21k+IN1k | 7.7 | 2.2 | 2.4 | 6.2 | 4.6 |
| YOLO11m-cls | IN1k | 16.0 | 4.0 | 4.5 | 15.5 | 10.0 |
| ViT-MAE | IN1k | 33.6 | 8.3 | 13.7 | 30.0 | 21.4 |
| DINOv1-ViT-B-16 | IN1k | 13.2 | 6.7 | 6.7 | 16.1 | 10.7 |
| PaliGemma3b | many | 11.5 | 2.5 | 3.9 | 12.2 | 7.5 |
| BEIT-B-16-224 | IN21k+IN1k | 10.1 | 1.4 | 1.8 | 6.7 | 5.0 |
| BEIT-L-16-224 | IN21k+IN1k | 6.5 | 1.0 | 1.0 | 4.6 | 3.3 |
| BEITv2-B-16-224-1K-1K | IN1k+IN1k | 7.7 | 1.2 | 1.5 | 5.8 | 4.1 |
| BEITv2-B-16-224-22K-1K | IN1k+(IN21k+IN1k) | 7.9 | 1.1 | 1.3 | 5.5 | 4.0 |
| SIGLIP-SO400-14-384 | WebLI | 8.5 | 1.8 | 3.3 | 8.9 | 5.6 |
| SIGLIP2-SO400-14-384 | WebLI | 6.3 | 1.4 | 2.0 | 4.5 | 3.6 |
| SIGLIP-B-16-224 | WebLI | 15.3 | 3.8 | 6.5 | 16.1 | 10.4 |
| SIGLIP2-B-16-224 | WebLI | 14.1 | 3.4 | 5.0 | 11.9 | 8.6 |

- The authors should discuss the computational efficiency of the proposed algorithms and how they scale with dataset size.

- If applicable, the authors should discuss possible limitations of their approach to address problems of privacy and fairness.

- While the authors might fear that complete honesty about limitations might be used by reviewers as grounds for rejection, a worse outcome might be that reviewers discover limitations that aren't acknowledged in the paper. The authors should use their best judgment and recognize that individual actions in favor of transparency play an important role in developing norms that preserve the integrity of the community. Reviewers will be specifically instructed to not penalize honesty concerning limitations.

3. **Theory assumptions and proofs**

Question: For each theoretical result, does the paper provide the full set of assumptions and a complete (and correct) proof?

Answer: [NA]

Justification:

Guidelines:

- The answer NA means that the paper does not include theoretical results.
- All the theorems, formulas, and proofs in the paper should be numbered and cross-referenced.
- All assumptions should be clearly stated or referenced in the statement of any theorems.
- The proofs can either appear in the main paper or the supplemental material, but if they appear in the supplemental material, the authors are encouraged to provide a short proof sketch to provide intuition.
- Inversely, any informal proof provided in the core of the paper should be complemented by formal proofs provided in appendix or supplemental material.

Table 9: Attack success rates (%) for GeometricMasksV1

| Model Name | Training Dataset | GeometricMasksV1 Circle 50 | GeometricMasksV1 Circle 80 | GeometricMasksV1 Circle 110 | GeometricMasksV1 Circle 140 | Avg. GeometricMasksV1 |
|---|---|---|---|---|---|---|
| ResNet50 | IN1k | 40.5 | 83.7 | 97.7 | 99.6 | 80.4 |
| ViT-B-16-224 | IN21k+IN1k | 15.1 | 29.4 | 55.0 | 83.4 | 45.7 |
| ViT-B-32-384 | IN21k+IN1k | 24.1 | 55.3 | 85.0 | 97.7 | 65.5 |
| ViT-L-16-384 | IN21k+IN1k | 16.8 | 39.0 | 69.0 | 91.1 | 54.0 |
| ViT-L-32-384 | IN21k+IN1k | 23.9 | 51.8 | 83.3 | 97.5 | 64.1 |
| CLIP-ViT-B-16 | DFN-2B | 18.3 | 39.7 | 65.5 | 86.6 | 52.5 |
| CLIP-ViT-B-32-QuickGELU | OPENAI-WIT | 36.0 | 62.1 | 83.2 | 95.3 | 69.2 |
| CLIP-ViT-B-32-QuickGELU | LAION-400M | 27.3 | 53.8 | 78.8 | 94.1 | 63.5 |
| CLIP-ViT-B-32 | LAION-2B | 27.9 | 55.2 | 80.9 | 95.1 | 64.8 |
| CLIP-EVA02-B-16 | LAION-2B(1.6B) + COYO-700M(400M) | 19.8 | 40.5 | 65.5 | 88.4 | 53.6 |
| CLIP-ViT-L-14-QuickGELU | MetaClip400M | 23.2 | 43.2 | 63.8 | 83.0 | 53.3 |
| CLIP-ViT-L-14-QuickGELU | OPENAI-WIT | 25.6 | 43.3 | 60.5 | 78.2 | 51.9 |
| CLIP-ViT-L-14 | DataComp-1B | 17.4 | 35.6 | 58.0 | 80.9 | 48.0 |
| CLIP-ViT-L-14 | MetaClip full CC | 18.8 | 35.3 | 55.6 | 77.3 | 46.7 |
| CLIP-ViT-L-14 | DFN-2B | 15.0 | 35.5 | 63.2 | 87.7 | 50.4 |
| CLIP-EVA02-L-14 | LAION-2B(1.6B) + COYO-700M(400M) | 11.6 | 24.9 | 44.9 | 69.8 | 37.8 |
| CLIP-ViT-H-14 | DFN-5B | 11.8 | 30.9 | 59.6 | 86.8 | 47.3 |
| DINOv2-S | LVD-142M | 20.2 | 47.9 | 77.7 | 95.0 | 60.2 |
| DINOv2-B | LVD-142M | 11.4 | 26.7 | 51.7 | 77.8 | 41.9 |
| DINOv2-L | LVD-142M | 5.7 | 11.6 | 23.8 | 43.6 | 21.2 |
| DINOv2-G | LVD-142M | 4.7 | 8.0 | 14.4 | 26.6 | 13.4 |
| SWIN-L-384 | IN1k | 8.2 | 11.7 | 19.5 | 32.5 | 18.0 |
| SWINv2-L-384 | IN21k+IN1k | 6.8 | 8.8 | 10.2 | 12.2 | 9.5 |
| ConvNext-L | IN21k+IN1k | 13.7 | 25.9 | 37.8 | 49.6 | 31.8 |
| YOLO11m-cls | IN1k | 50.4 | 90.8 | 98.9 | 99.7 | 85.0 |
| ViT-MAE | IN1k | 68.5 | 94.4 | 99.0 | 99.6 | 90.4 |
| DINOv1-ViT-B-16 | IN1k | 30.1 | 66.4 | 92.2 | 98.9 | 71.9 |
| PaliGemma3b | many | 26.1 | 59.5 | 84.7 | 96.5 | 66.7 |
| BEIT-B-16-224 | IN21k+IN1k | 12.7 | 29.9 | 56.9 | 83.2 | 45.7 |
| BEIT-L-16-224 | IN21k+IN1k | 8.9 | 20.5 | 41.8 | 71.0 | 35.5 |
| BEITv2-B-16-224-1K-1K | IN1k+IN1k | 11.0 | 19.7 | 34.5 | 58.4 | 30.9 |
| BEITv2-B-16-224-22K-1K | IN1k+(IN21k+IN1k) | 7.0 | 11.5 | 24.5 | 54.4 | 24.4 |
| SIGLIP-SO400-14-384 | WebLI | 13.0 | 28.1 | 48.7 | 71.7 | 40.4 |
| SIGLIP2-SO400-14-384 | WebLI | 11.8 | 24.0 | 42.0 | 65.9 | 35.9 |
| SIGLIP-B-16-224 | WebLI | 12.8 | 28.0 | 54.7 | 81.8 | 44.3 |
| SIGLIP2-B-16-224 | WebLI | 15.8 | 34.7 | 61.8 | 85.4 | 49.4 |

Table 10: Attack success rates (%) for GeometricMasksV2

| Model Name | Training Dataset | GeometricMasksV2 3-4-2 C1 Opacity 64 | GeometricMasksV2 3-4-2 C1 Opacity 96 | GeometricMasksV2 3-4-2 C1 Opacity 128 | GeometricMasksV2 3-4-5 C1 Opacity 128 | GeometricMasksV2 3-7-2 C1 Opacity 128 | GeometricMasksV2 6-4-2 C1 Opacity 128 | GeometricMasksV2 6-7-2 C1 Opacity 128 | Avg. GeometricMasksV2 |
|---|---|---|---|---|---|---|---|---|---|
| ResNet50 | IN1k | 33.7 | 62.1 | 92.2 | 87.7 | 99.1 | 99.0 | 99.6 | 81.9 |
| ViT-B-16-224 | IN21k+IN1k | 23.8 | 40.3 | 58.5 | 51.5 | 80.6 | 90.4 | 88.1 | 61.9 |
| ViT-B-32-384 | IN21k+IN1k | 16.3 | 33.8 | 58.6 | 53.2 | 94.5 | 88.1 | 98.4 | 63.3 |
| ViT-L-16-384 | IN21k+IN1k | 14.2 | 21.9 | 33.0 | 31.2 | 52.8 | 66.6 | 89.2 | 44.1 |
| ViT-L-32-384 | IN21k+IN1k | 18.5 | 34.5 | 57.7 | 52.1 | 89.0 | 85.4 | 98.5 | 62.2 |
| CLIP-ViT-B-16 | DFN-2B | 18.3 | 27.8 | 38.8 | 40.5 | 64.8 | 66.4 | 78.9 | 47.9 |
| CLIP-ViT-B-32-QuickGELU | OPENAI-WIT | 41.7 | 56.1 | 68.3 | 70.5 | 90.7 | 89.5 | 95.5 | 73.2 |
| CLIP-ViT-B-32-QuickGELU | LAION-400M | 39.0 | 58.4 | 80.4 | 76.9 | 93.3 | 96.2 | 92.8 | 76.7 |
| CLIP-ViT-B-32 | LAION-2B | 34.2 | 52.4 | 69.0 | 68.4 | 91.8 | 89.9 | 94.8 | 71.5 |
| CLIP-EVA02-B-16 | LAION-2B(1.6B) + COYO-700M(400M) | 20.5 | 29.7 | 40.0 | 44.4 | 69.0 | 71.0 | 83.5 | 51.2 |
| CLIP-ViT-L-14-QuickGELU | MetaClip400M | 17.2 | 22.7 | 28.8 | 30.8 | 43.1 | 48.8 | 69.0 | 37.2 |
| CLIP-ViT-L-14-QuickGELU | OPENAI-WIT | 18.9 | 25.5 | 32.4 | 36.4 | 43.7 | 46.3 | 68.9 | 38.9 |
| CLIP-ViT-L-14 | DataComp-1B | 15.8 | 22.4 | 28.7 | 29.3 | 42.3 | 44.6 | 67.7 | 35.8 |
| CLIP-ViT-L-14 | MetaClip full CC | 14.1 | 19.6 | 25.8 | 26.8 | 37.0 | 51.3 | 72.7 | 35.3 |
| CLIP-ViT-L-14 | DFN-2B | 11.2 | 18.4 | 27.3 | 28.4 | 47.5 | 53.2 | 79.0 | 37.9 |
| CLIP-EVA02-L-14 | LAION-2B(1.6B) + COYO-700M(400M) | 10.1 | 15.2 | 21.8 | 22.3 | 34.2 | 42.4 | 62.2 | 29.7 |
| CLIP-ViT-H-14 | DFN-5B | 8.9 | 15.3 | 24.2 | 24.1 | 52.6 | 64.8 | 87.4 | 39.6 |
| DINOv2-S | LVD-142M | 13.2 | 17.4 | 26.3 | 26.3 | 50.9 | 53.5 | 90.1 | 39.7 |
| DINOv2-B | LVD-142M | 7.5 | 9.3 | 12.6 | 12.7 | 22.9 | 28.2 | 60.2 | 21.9 |
| DINOv2-L | LVD-142M | 4.4 | 5.2 | 6.4 | 7.0 | 10.6 | 11.5 | 31.1 | 10.9 |
| DINOv2-G | LVD-142M | 3.8 | 4.6 | 5.8 | 6.1 | 8.6 | 9.8 | 18.3 | 8.1 |
| SWIN-L-384 | IN1k | 9.6 | 12.3 | 15.8 | 17.6 | 67.6 | 53.6 | 75.3 | 36.0 |
| SWINv2-L-384 | IN21k+IN1k | 5.8 | 6.7 | 7.9 | 8.2 | 25.8 | 17.6 | 47.2 | 17.0 |
| ConvNext-L | IN21k+IN1k | 8.2 | 11.4 | 15.9 | 20.6 | 18.7 | 30.2 | 43.2 | 21.2 |
| YOLO11m-cls | IN1k | 31.7 | 58.8 | 82.0 | 91.6 | 97.7 | 91.3 | 98.7 | 78.8 |
| ViT-MAE | IN1k | 58.8 | 81.2 | 92.8 | 94.9 | 98.7 | 98.0 | 99.2 | 89.1 |
| DINOv1-ViT-B-16 | IN1k | 29.5 | 46.7 | 63.0 | 63.2 | 90.6 | 92.4 | 98.9 | 69.2 |
| PaliGemma3b | many | 17.1 | 26.5 | 38.5 | 46.2 | 68.3 | 69.8 | 91.4 | 51.1 |
| BEIT-B-16-224 | IN21k+IN1k | 13.2 | 21.3 | 31.1 | 34.5 | 67.0 | 71.0 | 73.2 | 44.5 |
| BEIT-L-16-224 | IN21k+IN1k | 8.6 | 12.4 | 17.2 | 17.4 | 42.4 | 63.1 | 69.0 | 32.9 |
| BEITv2-B-16-224-1K-1K | IN1k+IN1k | 8.8 | 12.9 | 17.9 | 19.6 | 32.4 | 77.9 | 68.7 | 34.0 |
| BEITv2-B-16-224-22K-1K | IN1k+(IN21k+IN1k) | 5.7 | 9.4 | 20.7 | 18.5 | 29.8 | 49.9 | 69.0 | 29.0 |
| SIGLIP-SO400-14-384 | WebLI | 7.9 | 11.8 | 16.5 | 18.9 | 27.9 | 28.8 | 50.4 | 23.2 |
| SIGLIP2-SO400-14-384 | WebLI | 9.3 | 12.1 | 16.0 | 16.2 | 24.6 | 22.0 | 43.1 | 20.5 |
| SIGLIP-B-16-224 | WebLI | 22.2 | 34.1 | 47.7 | 48.1 | 79.1 | 73.7 | 83.1 | 55.5 |
| SIGLIP2-B-16-224 | WebLI | 23.1 | 35.1 | 47.7 | 46.6 | 76.6 | 74.5 | 79.9 | 54.8 |

- Theorems and Lemmas that the proof relies upon should be properly referenced.

4. **Experimental result reproducibility**

    Question: Does the paper fully disclose all the information needed to reproduce the main experimental results of the paper to the extent that it affects the main claims and/or conclusions of the paper (regardless of whether the code and data are provided or not)?

    Answer: [Yes]

    Justification: We run the models using the standard setups and will provide the codebase.

    Guidelines:

    - The answer NA means that the paper does not include experiments.
    - If the paper includes experiments, a No answer to this question will not be perceived well by the reviewers: Making the paper reproducible is important, regardless of whether the code and data are provided or not.
    - If the contribution is a dataset and/or model, the authors should describe the steps taken to make their results reproducible or verifiable.

Table 11: Attack success rates (%) for COCO Objects

| Model Name | Training Dataset | COCO Objects Black Background | COCO Objects Thresholded Perlin Noise Background | COCO Objects Perlin Noise Background | Avg. COCO Objects |
|---|---|---|---|---|---|
| ResNet50 | IN1k | 18.3 | 18.6 | 22.3 | 19.7 |
| ViT-B-16-224 | IN21k+IN1k | 24.3 | 25.5 | 36.0 | 28.6 |
| ViT-B-32-384 | IN21k+IN1k | 21.0 | 23.8 | 32.7 | 25.8 |
| ViT-L-16-384 | IN21k+IN1k | 15.0 | 17.6 | 29.2 | 20.6 |
| ViT-L-32-384 | IN21k+IN1k | 20.8 | 24.4 | 35.7 | 27.0 |
| CLIP-VIT-B-16 | DFN-2B | 20.5 | 22.8 | 27.6 | 23.7 |
| CLIP-VIT-B-32-QuickGELU | OPENAI-WIT | 28.5 | 37.8 | 41.2 | 35.8 |
| CLIP-VIT-B-32-QuickGELU | LAION-400M | 30.3 | 36.7 | 41.6 | 36.2 |
| CLIP-VIT-B-32 | LAION-2B | 29.4 | 33.8 | 36.8 | 33.3 |
| CLIP-EVA02-B-16 | LAION-2B(1.6B) + COYO-700M(400M) | 18.6 | 19.9 | 23.4 | 20.6 |
| CLIP-VIT-L-14-QuickGELU | MetaClip400M | 20.5 | 22.0 | 27.5 | 23.4 |
| CLIP-VIT-L-14-QuickGELU | OPENAI-WIT | 16.5 | 22.6 | 28.8 | 22.6 |
| CLIP-VIT-L-14 | DataComp-1B | 16.0 | 17.9 | 20.2 | 18.0 |
| CLIP-VIT-L-14 | MetaClip full CC | 19.0 | 21.0 | 24.0 | 21.3 |
| CLIP-VIT-L-14 | DFN-2B | 16.4 | 17.9 | 21.4 | 18.6 |
| CLIP-EVA02-L-14 | LAION-2B(1.6B) + COYO-700M(400M) | 13.6 | 15.8 | 17.2 | 15.5 |
| CLIP-VIT-H-14 | DFN-5B | 14.4 | 16.2 | 20.9 | 17.2 |
| DINOv2-S | LVD-142M | 17.4 | 21.3 | 26.2 | 21.6 |
| DINOv2-B | LVD-142M | 12.9 | 17.5 | 17.6 | 16.0 |
| DINOv2-L | LVD-142M | 9.4 | 12.0 | 10.7 | 10.7 |
| DINOv2-G | LVD-142M | 8.2 | 13.2 | 10.8 | 10.7 |
| SWIN-L-384 | IN1k | 8.6 | 26.8 | 41.5 | 25.6 |
| SWINv2-L-384 | IN21k+IN1k | 8.1 | 30.1 | 36.4 | 24.9 |
| ConvNext-L | IN21k+IN1k | 12.1 | 12.7 | 16.9 | 13.9 |
| YOLO11m-cls | IN1k | 21.7 | 30.0 | 38.9 | 30.2 |
| ViT-MAE | IN1k | 46.5 | 54.6 | 80.6 | 60.5 |
| DINOv1-ViT-B-16 | IN1k | 19.5 | 22.8 | 32.2 | 24.8 |
| PaliGemma3b | many | 16.2 | 19.3 | 23.0 | 19.5 |
| BEIT-B-16-224 | IN21k+IN1k | 16.7 | 17.4 | 27.8 | 20.7 |
| BEIT-L-16-224 | IN21k+IN1k | 11.4 | 13.2 | 21.2 | 15.3 |
| BEITv2-B-16-224-1K-1K | IN1k+IN1k | 9.8 | 15.1 | 61.1 | 28.7 |
| BEITv2-B-16-224-22K-1K | IN1k+(IN21k+IN1k) | 10.8 | 11.8 | 23.9 | 15.5 |
| SIGLIP-SO400-14-384 | WebLI | 12.0 | 14.7 | 18.2 | 15.0 |
| SIGLIP2-SO400-14-384 | WebLI | 9.8 | 13.4 | 14.7 | 12.6 |
| SIGLIP-B-16-224 | WebLI | 23.9 | 26.0 | 32.0 | 27.3 |
| SIGLIP2-B-16-224 | WebLI | 21.1 | 25.1 | 28.4 | 24.8 |

- Depending on the contribution, reproducibility can be accomplished in various ways. For example, if the contribution is a novel architecture, describing the architecture fully might suffice, or if the contribution is a specific model and empirical evaluation, it may be necessary to either make it possible for others to replicate the model with the same dataset, or provide access to the model. In general. releasing code and data is often one good way to accomplish this, but reproducibility can also be provided via detailed instructions for how to replicate the results, access to a hosted model (e.g., in the case of a large language model), releasing of a model checkpoint, or other means that are appropriate to the research performed.

- While NeurIPS does not require releasing code, the conference does require all submissions to provide some reasonable avenue for reproducibility, which may depend on the nature of the contribution. For example

  (a) If the contribution is primarily a new algorithm, the paper should make it clear how to reproduce that algorithm.

  (b) If the contribution is primarily a new model architecture, the paper should describe the architecture clearly and fully.

  (c) If the contribution is a new model (e.g., a large language model), then there should either be a way to access this model for reproducing the results or a way to reproduce the model (e.g., with an open-source dataset or instructions for how to construct the dataset).

  (d) We recognize that reproducibility may be tricky in some cases, in which case authors are welcome to describe the particular way they provide for reproducibility. In the case of closed-source models, it may be that access to the model is limited in

Table 12: Attack success rates (%) for ImageNet-C approximated using ImageNet-1K

| Model Name | Training Dataset | ImageNet-C Distortion Severity 1 | ImageNet-C Distortion Severity 2 | ImageNet-C Distortion Severity 3 | ImageNet-C Distortion Severity 4 | ImageNet-C Distortion Severity 5 | Avg. ImageNet-C |
|---|---|---|---|---|---|---|---|
| ResNet50 | IN1k | 16.4 | 26.9 | 36.9 | 50.9 | 65.4 | 39.3 |
| ViT-B-16-224 | IN21k+IN1k | 7.1 | 14.4 | 20.8 | 33.0 | 50.1 | 25.1 |
| ViT-B-32-384 | IN21k+IN1k | 8.7 | 16.1 | 22.2 | 35.3 | 52.1 | 26.9 |
| ViT-L-16-384 | IN21k+IN1k | 7.7 | 13.1 | 17.7 | 26.8 | 41.3 | 21.3 |
| ViT-L-32-384 | IN21k+IN1k | 7.7 | 14.3 | 19.5 | 30.1 | 46.1 | 23.5 |
| CLIP-VIT-B-16 | DFN-2B | 12.1 | 22.5 | 30.8 | 43.3 | 57.7 | 33.3 |
| CLIP-VIT-B-32-QuickGELU | OPENAI-WIT | 12.8 | 23.1 | 32.8 | 47.0 | 62.4 | 35.6 |
| CLIP-VIT-B-32-QuickGELU | LAION-400M | 15.5 | 26.6 | 35.9 | 49.7 | 64.2 | 38.4 |
| CLIP-VIT-B-32 | LAION-2B | 15.0 | 25.8 | 35.0 | 48.4 | 63.1 | 37.5 |
| CLIP-EVA02-B-16 | LAION-2B(1.6B) + COYO-700M(400M) | 8.2 | 16.1 | 23.6 | 34.7 | 49.7 | 26.5 |
| CLIP-VIT-L-14-QuickGELU | MetaClip400M | 10.4 | 18.7 | 25.8 | 37.9 | 51.6 | 28.9 |
| CLIP-VIT-L-14-QuickGELU | OPENAI-WIT | 8.7 | 15.6 | 21.9 | 32.1 | 45.5 | 24.8 |
| CLIP-VIT-L-14 | DataComp-1B | 7.2 | 13.2 | 18.8 | 28.3 | 40.8 | 21.7 |
| CLIP-VIT-L-14 | MetaClip full CC | 8.5 | 16.1 | 22.0 | 32.4 | 45.5 | 24.9 |
| CLIP-VIT-L-14 | DFN-2B | 6.7 | 13.5 | 19.3 | 28.7 | 41.6 | 22.0 |
| CLIP-EVA02-L-14 | LAION-2B(1.6B) + COYO-700M(400M) | 5.4 | 9.4 | 13.2 | 19.4 | 29.6 | 15.4 |
| CLIP-VIT-H-14 | DFN-5B | 6.4 | 12.2 | 17.5 | 26.0 | 39.3 | 20.3 |
| DINOv2-S | LVD-142M | 9.2 | 17.2 | 24.9 | 37.0 | 53.3 | 28.3 |
| DINOv2-B | LVD-142M | 6.0 | 11.1 | 16.2 | 24.9 | 38.4 | 19.3 |
| DINOv2-L | LVD-142M | 3.6 | 6.7 | 9.2 | 14.2 | 23.4 | 11.4 |
| DINOv2-G | LVD-142M | 3.3 | 6.1 | 7.9 | 11.7 | 19.0 | 9.6 |
| SWIN-L-384 | IN1k | 7.6 | 11.1 | 15.5 | 21.3 | 31.5 | 17.4 |
| SWINv2-L-384 | IN21k+IN1k | 7.1 | 11.0 | 14.6 | 19.7 | 29.4 | 16.4 |
| ConvNext-L | IN21k+IN1k | 8.2 | 13.4 | 17.5 | 24.7 | 35.5 | 19.9 |
| YOLO11m-cls | IN1k | 25.4 | 39.6 | 51.7 | 65.9 | 77.8 | 52.1 |
| ViT-MAE | IN1k | 30.3 | 45.5 | 56.9 | 69.9 | 81.7 | 56.9 |
| DINOv1-ViT-B-16 | IN1k | 11.7 | 21.9 | 32.7 | 47.4 | 63.8 | 35.5 |
| PaliGemma3b | many | 11.0 | 20.5 | 29.6 | 42.5 | 57.5 | 32.2 |
| BEIT-B-16-224 | IN21k+IN1k | 6.5 | 11.3 | 15.9 | 23.9 | 37.0 | 18.9 |
| BEIT-L-16-224 | IN21k+IN1k | 4.6 | 7.9 | 10.5 | 15.6 | 25.4 | 12.8 |
| BEITv2-B-16-224-1K-1K | IN1k+IN1k | 7.7 | 12.1 | 17.4 | 24.6 | 36.2 | 19.6 |
| BEITv2-B-16-224-22K-1K | IN1k+(IN21k+IN1k) | 6.6 | 11.2 | 14.5 | 20.7 | 31.1 | 16.8 |
| SIGLIP-SO400-14-384 | WebLI | 7.7 | 14.4 | 20.9 | 31.3 | 44.7 | 23.8 |
| SIGLIP2-SO400-14-384 | WebLI | 4.3 | 8.7 | 13.8 | 22.3 | 34.6 | 16.7 |
| SIGLIP-B-16-224 | WebLI | 14.0 | 25.8 | 36.0 | 48.8 | 63.1 | 37.5 |
| SIGLIP2-B-16-224 | WebLI | 11.0 | 21.7 | 30.5 | 42.7 | 56.7 | 32.5 |

some way (e.g., to registered users), but it should be possible for other researchers to have some path to reproducing or verifying the results.

5. **Open access to data and code**

Question: Does the paper provide open access to the data and code, with sufficient instructions to faithfully reproduce the main experimental results, as described in supplemental material?

Answer: [Yes]

Justification: The code is provided in a zip file

Guidelines:

- The answer NA means that paper does not include experiments requiring code.
- Please see the NeurIPS code and data submission guidelines (`https://nips.cc/public/guides/CodeSubmissionPolicy`) for more details.
- While we encourage the release of code and data, we understand that this might not be possible, so "No" is an acceptable answer. Papers cannot be rejected simply for not including code, unless this is central to the contribution (e.g., for a new open-source benchmark).
- The instructions should contain the exact command and environment needed to run to reproduce the results. See the NeurIPS code and data submission guidelines (`https://nips.cc/public/guides/CodeSubmissionPolicy`) for more details.
- The authors should provide instructions on data access and preparation, including how to access the raw data, preprocessed data, intermediate data, and generated data, etc.
- The authors should provide scripts to reproduce all experimental results for the new proposed method and baselines. If only a subset of experiments are reproducible, they should state which ones are omitted from the script and why.
- At submission time, to preserve anonymity, the authors should release anonymized versions (if applicable).
- Providing as much information as possible in supplemental material (appended to the paper) is recommended, but including URLs to data and code is permitted.

Table 13: Attack success rates (%) for ImageNet-R approximated using ImageNet-200

| Model Name | Training Dataset | ImageNet-R |
|---|---|---|
| ResNet50 | IN1k | 56.5 |
| ViT-B-16-224 | IN21k+IN1k | 50.2 |
| ViT-B-32-384 | IN21k+IN1k | 52.7 |
| ViT-L-16-384 | IN21k+IN1k | 42.6 |
| ViT-L-32-384 | IN21k+IN1k | 49.3 |
| CLIP-VIT-B-16 | DFN-2B | 15.3 |
| CLIP-VIT-B-32-QuickGELU | OPENAI-WIT | 26.7 |
| CLIP-VIT-B-32-QuickGELU | LAION-400M | 20.4 |
| CLIP-VIT-B-32 | LAION-2B | 19.5 |
| CLIP-EVA02-B-16 | LAION-2B(1.6B) + COYO-700M(400M) | 17.0 |
| CLIP-VIT-L-14-QuickGELU | MetaClip400M | 9.5 |
| CLIP-VIT-L-14-QuickGELU | OPENAI-WIT | 11.0 |
| CLIP-VIT-L-14 | DataComp-1B | 7.8 |
| CLIP-VIT-L-14 | MetaClip full CC | 8.3 |
| CLIP-VIT-L-14 | DFN-2B | 9.0 |
| CLIP-EVA02-L-14 | LAION-2B(1.6B) + COYO-700M(400M) | 6.3 |
| CLIP-VIT-H-14 | DFN-5B | 6.9 |
| DINOv2-S | LVD-142M | 40.7 |
| DINOv2-B | LVD-142M | 30.0 |
| DINOv2-L | LVD-142M | 19.4 |
| DINOv2-G | LVD-142M | 17.9 |
| SWIN-L-384 | IN1k | 31.2 |
| SWINv2-L-384 | IN21k+IN1k | 29.7 |
| ConvNext-L | IN21k+IN1k | 32.6 |
| YOLO11m-cls | IN1k | 57.4 |
| ViT-MAE | IN1k | 76.3 |
| DINOv1-ViT-B-16 | IN1k | 61.6 |
| PaliGemma3b | many | 15.1 |
| BEIT-B-16-224 | IN21k+IN1k | 36.6 |
| BEIT-L-16-224 | IN21k+IN1k | 26.3 |
| BEITv2-B-16-224-1K-1K | IN1k+IN1k | 35.2 |
| BEITv2-B-16-224-22K-1K | IN1k+(IN21k+IN1k) | 30.8 |
| SIGLIP-SO400-14-384 | WebLI | 5.1 |
| SIGLIP2-SO400-14-384 | WebLI | 4.0 |
| SIGLIP-B-16-224 | WebLI | 8.0 |
| SIGLIP2-B-16-224 | WebLI | 8.8 |

6. **Experimental setting/details**

Question: Does the paper specify all the training and test details (e.g., data splits, hyper-parameters, how they are chosen, type of optimizer, etc.) necessary to understand the results?

Answer: [Yes]

Justification:

Guidelines:

- The answer NA means that the paper does not include experiments.
- The experimental setting should be presented in the core of the paper to a level of detail that is necessary to appreciate the results and make sense of them.
- The full details can be provided either with the code, in appendix, or as supplemental material.

7. **Experiment statistical significance**

Question: Does the paper report error bars suitably and correctly defined or other appropriate information about the statistical significance of the experiments?

Answer: [No]

Justification:

Guidelines:

- The answer NA means that the paper does not include experiments.
- The authors should answer "Yes" if the results are accompanied by error bars, confidence intervals, or statistical significance tests, at least for the experiments that support the main claims of the paper.
- The factors of variability that the error bars are capturing should be clearly stated (for example, train/test split, initialization, random drawing of some parameter, or overall run with given experimental conditions).
- The method for calculating the error bars should be explained (closed form formula, call to a library function, bootstrap, etc.)
- The assumptions made should be given (e.g., Normally distributed errors).
- It should be clear whether the error bar is the standard deviation or the standard error of the mean.
- It is OK to report 1-sigma error bars, but one should state it. The authors should preferably report a 2-sigma error bar than state that they have a 96% CI, if the hypothesis of Normality of errors is not verified.
- For asymmetric distributions, the authors should be careful not to show in tables or figures symmetric error bars that would yield results that are out of range (e.g. negative error rates).
- If error bars are reported in tables or plots, The authors should explain in the text how they are calculated and reference the corresponding figures or tables in the text.

8. **Experiments compute resources**

Question: For each experiment, does the paper provide sufficient information on the computer resources (type of compute workers, memory, time of execution) needed to reproduce the experiments?

Answer: [Yes]

Justification: See overall numbers in Appendix I.

Guidelines:

- The answer NA means that the paper does not include experiments.
- The paper should indicate the type of compute workers CPU or GPU, internal cluster, or cloud provider, including relevant memory and storage.
- The paper should provide the amount of compute required for each of the individual experimental runs as well as estimate the total compute.
- The paper should disclose whether the full research project required more compute than the experiments reported in the paper (e.g., preliminary or failed experiments that didn't make it into the paper).

9. **Code of ethics**

Question: Does the research conducted in the paper conform, in every respect, with the NeurIPS Code of Ethics `https://neurips.cc/public/EthicsGuidelines`?

Answer: [Yes]

Justification:

Guidelines:

- The answer NA means that the authors have not reviewed the NeurIPS Code of Ethics.
- If the authors answer No, they should explain the special circumstances that require a deviation from the Code of Ethics.

- The authors should make sure to preserve anonymity (e.g., if there is a special consideration due to laws or regulations in their jurisdiction).

10. **Broader impacts**

Question: Does the paper discuss both potential positive societal impacts and negative societal impacts of the work performed?

Answer: [NA]

Justification: There are no direct impacts from this work.

Guidelines:

- The answer NA means that there is no societal impact of the work performed.
- If the authors answer NA or No, they should explain why their work has no societal impact or why the paper does not address societal impact.
- Examples of negative societal impacts include potential malicious or unintended uses (e.g., disinformation, generating fake profiles, surveillance), fairness considerations (e.g., deployment of technologies that could make decisions that unfairly impact specific groups), privacy considerations, and security considerations.
- The conference expects that many papers will be foundational research and not tied to particular applications, let alone deployments. However, if there is a direct path to any negative applications, the authors should point it out. For example, it is legitimate to point out that an improvement in the quality of generative models could be used to generate deepfakes for disinformation. On the other hand, it is not needed to point out that a generic algorithm for optimizing neural networks could enable people to train models that generate Deepfakes faster.
- The authors should consider possible harms that could arise when the technology is being used as intended and functioning correctly, harms that could arise when the technology is being used as intended but gives incorrect results, and harms following from (intentional or unintentional) misuse of the technology.
- If there are negative societal impacts, the authors could also discuss possible mitigation strategies (e.g., gated release of models, providing defenses in addition to attacks, mechanisms for monitoring misuse, mechanisms to monitor how a system learns from feedback over time, improving the efficiency and accessibility of ML).

11. **Safeguards**

Question: Does the paper describe safeguards that have been put in place for responsible release of data or models that have a high risk for misuse (e.g., pretrained language models, image generators, or scraped datasets)?

Answer: [NA]

Justification:

Guidelines:

- The answer NA means that the paper poses no such risks.
- Released models that have a high risk for misuse or dual-use should be released with necessary safeguards to allow for controlled use of the model, for example by requiring that users adhere to usage guidelines or restrictions to access the model or implementing safety filters.
- Datasets that have been scraped from the Internet could pose safety risks. The authors should describe how they avoided releasing unsafe images.
- We recognize that providing effective safeguards is challenging, and many papers do not require this, but we encourage authors to take this into account and make a best faith effort.

12. **Licenses for existing assets**

Question: Are the creators or original owners of assets (e.g., code, data, models), used in the paper, properly credited and are the license and terms of use explicitly mentioned and properly respected?

Answer: [Yes]

Justification:

Guidelines:

- The answer NA means that the paper does not use existing assets.
- The authors should cite the original paper that produced the code package or dataset.
- The authors should state which version of the asset is used and, if possible, include a URL.
- The name of the license (e.g., CC-BY 4.0) should be included for each asset.
- For scraped data from a particular source (e.g., website), the copyright and terms of service of that source should be provided.
- If assets are released, the license, copyright information, and terms of use in the package should be provided. For popular datasets, `paperswithcode.com/datasets` has curated licenses for some datasets. Their licensing guide can help determine the license of a dataset.
- For existing datasets that are re-packaged, both the original license and the license of the derived asset (if it has changed) should be provided.
- If this information is not available online, the authors are encouraged to reach out to the asset's creators.

13. **New assets**

Question: Are new assets introduced in the paper well documented and is the documentation provided alongside the assets?

Answer: [NA]

Justification:

Guidelines:

- The answer NA means that the paper does not release new assets.
- Researchers should communicate the details of the dataset/code/model as part of their submissions via structured templates. This includes details about training, license, limitations, etc.
- The paper should discuss whether and how consent is obtained from people whose asset is used.
- At submission time, remember to anonymize your assets (if applicable). You can either create an anonymized URL or include an anonymized zip file.

14. **Crowdsourcing and research with human subjects**

Question: For crowdsourcing experiments and research with human subjects, does the paper include the full text of instructions given to participants and screenshots, if applicable, as well as details about compensation (if any)?

Answer: [Yes]

Justification: See Figure 15. Human volunteers are used for a small-scale experiment to annotate images and validate that the masks by Jabary et al. [30] are semantic preserving.

Guidelines:

- The answer NA means that the paper does not involve crowdsourcing nor research with human subjects.
- Including this information in the supplemental material is fine, but if the main contribution of the paper involves human subjects, then as much detail as possible should be included in the main paper.
- According to the NeurIPS Code of Ethics, workers involved in data collection, curation, or other labor should be paid at least the minimum wage in the country of the data collector.

15. **Institutional review board (IRB) approvals or equivalent for research with human subjects**

Question: Does the paper describe potential risks incurred by study participants, whether such risks are disclosed to the subjects, and whether Institutional Review Board (IRB) approvals (or an equivalent approval/review based on the requirements of your country or institution) are obtained?

Answer: [NA]

Justification:

Guidelines:

- The answer NA means that the paper does not involve crowdsourcing nor research with human subjects.
- Depending on the country in which research is conducted, IRB approval (or equivalent) may be required for any human subjects research. If you obtained IRB approval, you should clearly state this in the paper.
- We recognize that the procedures for this may vary significantly between institutions and locations, and we expect authors to adhere to the NeurIPS Code of Ethics and the guidelines for their institution.
- For initial submissions, do not include any information that would break anonymity (if applicable), such as the institution conducting the review.

16. **Declaration of LLM usage**

Question: Does the paper describe the usage of LLMs if it is an important, original, or non-standard component of the core methods in this research? Note that if the LLM is used only for writing, editing, or formatting purposes and does not impact the core methodology, scientific rigorousness, or originality of the research, declaration is not required.

Answer: [NA]

Justification:

Guidelines:

- The answer NA means that the core method development in this research does not involve LLMs as any important, original, or non-standard components.
- Please refer to our LLM policy (`https://neurips.cc/Conferences/2025/LLM`) for what should or should not be described.

Table 14: Average attack success rates (%) of models on the different attacks classes.

| Model Name | Training Dataset | Training Procedure | Training Data Size (M) | Random Perturbations | GeometricMasksV1 | GeometricMasksV2 | COCO Objects | ImageNet-C | Overall |
|---|---|---|---|---|---|---|---|---|---|
| ResNet50 | IN1k | supervised | 1.2 | 7.4 | 80.4 | 81.9 | 19.7 | 39.3 | 45.8 |
| ViT-B-16-224 | IN21k+IN1k | supervised | 14.0 | 11.6 | 45.7 | 61.9 | 28.6 | 25.1 | 34.6 |
| ViT-B-32-384 | IN21k+IN1k | supervised | 14.0 | 10.3 | 65.5 | 63.3 | 25.8 | 26.9 | 38.4 |
| ViT-L-16-384 | IN21k+IN1k | supervised | 14.0 | 8.0 | 54.0 | 44.1 | 20.6 | 21.3 | 29.6 |
| ViT-L-32-384 | IN21k+IN1k | supervised | 14.0 | 12.0 | 64.1 | 62.2 | 27.0 | 23.5 | 37.8 |
| CLIP-ViT-B-16 | DFN-2B | contrastive | 2000.0 | 9.4 | 52.5 | 47.9 | 23.7 | 33.3 | 33.4 |
| CLIP-ViT-B-32-QuickGELU | OPENAI-WIT | contrastive | 400.0 | 16.3 | 69.2 | 73.2 | 35.8 | 35.6 | 46.0 |
| CLIP-ViT-B-32-QuickGELU | LAION-400M | contrastive | 413.0 | 16.0 | 63.5 | 76.7 | 36.2 | 38.4 | 46.2 |
| CLIP-ViT-B-32 | LAION-2B | contrastive | 2000.0 | 14.4 | 64.8 | 71.5 | 33.3 | 37.5 | 44.3 |
| CLIP-EVA02-B-16 | LAION-2B(1.6B) + COYO-700M(400M) | contrastive | 2000.0 | 9.3 | 53.6 | 51.2 | 20.6 | 26.5 | 32.2 |
| CLIP-ViT-L-14-QuickGELU | MetaClip400M | contrastive | 400.0 | 10.3 | 53.3 | 37.2 | 23.4 | 28.9 | 30.6 |
| CLIP-ViT-L-14-QuickGELU | OPENAI-WIT | contrastive | 400.0 | 9.6 | 51.9 | 38.9 | 22.6 | 24.8 | 29.5 |
| CLIP-ViT-L-14 | DataComp-1B | contrastive | 1400.0 | 7.5 | 48.0 | 35.8 | 18.0 | 21.7 | 26.2 |
| CLIP-ViT-L-14 | MetaClip full CC | contrastive | 2500.0 | 9.3 | 46.7 | 35.3 | 21.3 | 24.9 | 27.5 |
| CLIP-ViT-L-14 | DFN-2B | contrastive | 2000.0 | 6.5 | 50.4 | 37.9 | 18.6 | 22.0 | 27.0 |
| CLIP-EVA02-L-14 | LAION-2B(1.6B) + COYO-700M(400M) | contrastive | 2000.0 | 6.5 | 37.8 | 29.7 | 15.5 | 15.4 | 21.0 |
| CLIP-ViT-H-14 | DFN-5B | contrastive | 5000.0 | 5.5 | 47.3 | 39.6 | 17.2 | 20.3 | 26.0 |
| DINOv2-S | LVD-142M | self-supervised | 142.0 | 4.9 | 60.2 | 39.7 | 21.6 | 28.3 | 31.0 |
| DINOv2-B | LVD-142M | self-supervised | 142.0 | 3.2 | 41.9 | 21.9 | 16.0 | 19.3 | 20.5 |
| DINOv2-L | LVD-142M | self-supervised | 142.0 | 2.0 | 21.2 | 10.9 | 10.7 | 11.4 | 11.3 |
| DINOv2-G | LVD-142M | self-supervised | 142.0 | 1.9 | 13.4 | 8.1 | 10.7 | 9.6 | 8.7 |
| SWIN-L-384 | IN1k | supervised | 1.2 | 3.6 | 18.0 | 36.0 | 25.6 | 17.4 | 20.1 |
| SWINv2-L-384 | IN21k+IN1k | supervised | 14.0 | 3.2 | 9.5 | 17.0 | 24.9 | 16.4 | 14.2 |
| ConvNext-L | IN21k+IN1k | supervised | 14.0 | 4.6 | 31.8 | 21.2 | 13.9 | 19.9 | 18.3 |
| YOLO11m-cls | IN1k | supervised | 1.2 | 10.0 | 85.0 | 78.8 | 30.2 | 52.1 | 51.2 |
| ViT-MAE | IN1k | self-supervised | 1.2 | 21.4 | 90.4 | 89.1 | 60.5 | 56.9 | 63.7 |
| DINOv1-ViT-B-16 | IN1k | self-supervised | 1.2 | 10.7 | 71.9 | 69.2 | 24.8 | 35.5 | 42.4 |
| PaliGemma3b | many | self-supervised | 22256.0 | 7.5 | 66.7 | 51.1 | 19.5 | 32.2 | 35.4 |
| BEIT-B-16-224 | IN21k+IN1k | self-supervised | 14.0 | 5.0 | 45.7 | 44.5 | 20.7 | 18.9 | 26.9 |
| BEIT-L-16-224 | IN21k+IN1k | self-supervised | 14.0 | 3.3 | 35.5 | 32.9 | 15.3 | 12.8 | 20.0 |
| BEITv2-B-16-224-1K-1K | IN1k+IN1k | self-supervised | 1.2 | 4.1 | 30.9 | 34.0 | 28.7 | 19.6 | 23.5 |
| BEITv2-B-16-224-22K-1K | IN1k+(IN21k+IN1k) | self-supervised | 14.0 | 4.0 | 24.4 | 29.0 | 15.5 | 16.8 | 17.9 |
| SIGLIP-SO400M-14-384 | WebLI | contrastive | 22000.0 | 5.6 | 40.4 | 23.2 | 15.0 | 23.8 | 21.6 |
| SIGLIP2-SO400M-14-384 | WebLI | contrastive | 22000.0 | 3.6 | 35.9 | 20.5 | 12.6 | 16.7 | 17.9 |
| SIGLIP-B-16-224 | WebLI | contrastive | 22000.0 | 10.4 | 44.3 | 55.5 | 27.3 | 37.5 | 35.0 |
| SIGLIP2-B-16-224 | WebLI | contrastive | 22000.0 | 8.6 | 49.4 | 54.8 | 24.8 | 32.5 | 34.0 |
| NaN | NaN | NaN | 3652.0 | 8.0 | 49.0 | 45.2 | 23.0 | 26.2 | 30.3 |

Table 15: Raw attack success rates (%). QG is QuickGELU, Training procedures are supervised (S), self-supervised (SS), and contrastive (C).

| Model Name | Training Dataset | Training Procedure | Training Dataset Size (M) | Random Hue 0.5 | Random Saturation 0.9 | Random Contrast 0.9 | Random Brightness 0.7 | Circle 50 | Circle 80 | Circle 110 | Circle 140 | 3-4-2 C1 Opacity 64 | 3-4-2 C1 Opacity 96 | 3-4-2 C1 Opacity 128 | 3-4-5 C1 Opacity 128 | 3-7-2 C1 Opacity 128 | 6-4-2 C1 Opacity 128 | 6-7-2 C1 Opacity 128 | Black Background | Thresholded Perlin Noise Background | Perlin Noise Background | ImageNet-R | IN-C 1 | IN-C 2 | IN-C 3 | IN-C 4 | IN-C 5 |
|---|---|---|---|---|---|---|---|---|---|---|---|---|---|---|---|---|---|---|---|---|---|---|---|---|---|---|---|
| ResNet50 | IN1k | S | 1.2 | 14.6 | 2.3 | 2.8 | 10.0 | 40.5 | 83.7 | 97.7 | 99.6 | 33.7 | 62.1 | 92.2 | 87.7 | 99.1 | 99.0 | 99.6 | 18.3 | 18.6 | 22.3 | 56.5 | 16.4 | 26.9 | 36.9 | 50.9 | 65.4 |
| ViT-B-16-224 | IN21k+IN1k | S | 14.0 | 18.8 | 4.2 | 6.6 | 16.7 | 15.1 | 29.4 | 55.0 | 83.4 | 23.8 | 40.3 | 58.5 | 51.5 | 80.6 | 90.4 | 88.1 | 24.3 | 25.5 | 36.0 | 50.2 | 7.1 | 14.4 | 20.8 | 33.0 | 50.1 |
| ViT-B-32-384 | IN21k+IN1k | S | 14.0 | 16.8 | 3.7 | 5.5 | 15.4 | 24.1 | 55.3 | 85.0 | 97.7 | 16.3 | 33.8 | 58.6 | 53.2 | 94.5 | 88.1 | 98.4 | 21.0 | 23.8 | 32.7 | 52.7 | 8.7 | 16.1 | 22.2 | 35.3 | 52.1 |
| ViT-L-16-384 | IN21k+IN1k | S | 14.0 | 12.8 | 2.8 | 4.6 | 11.8 | 16.8 | 39.0 | 69.0 | 91.1 | 14.2 | 21.9 | 33.0 | 31.2 | 52.8 | 66.6 | 89.2 | 15.0 | 17.6 | 29.2 | 42.6 | 7.7 | 13.1 | 17.7 | 26.8 | 41.3 |
| ViT-L-32-384 | IN21k+IN1k | S | 14.0 | 19.4 | 4.7 | 7.4 | 16.5 | 23.9 | 51.8 | 83.3 | 97.5 | 18.5 | 34.5 | 57.7 | 52.1 | 89.0 | 85.4 | 98.5 | 20.8 | 24.4 | 35.7 | 49.3 | 7.7 | 14.3 | 19.5 | 30.1 | 46.1 |
| CLIP-ViT-B-16 | DFN-2B | C | 2000.0 | 16.0 | 3.6 | 4.3 | 13.8 | 18.3 | 39.7 | 65.5 | 86.6 | 18.3 | 27.8 | 38.8 | 40.5 | 64.8 | 66.4 | 78.9 | 20.5 | 22.8 | 27.6 | 15.3 | 12.1 | 22.5 | 30.8 | 43.3 | 57.7 |
| CLIP-ViT-B-32-QG | OPENAI-WIT | C | 400.0 | 27.5 | 7.6 | 9.4 | 20.8 | 36.0 | 62.1 | 83.2 | 95.3 | 41.7 | 56.1 | 68.3 | 70.5 | 90.7 | 89.5 | 95.5 | 28.5 | 37.8 | 41.2 | 26.7 | 12.8 | 23.1 | 32.8 | 47.0 | 62.4 |
| CLIP-ViT-B-32-QG | LAION-400M | C | 413.0 | 25.8 | 7.1 | 9.4 | 21.8 | 27.3 | 53.8 | 78.8 | 94.1 | 39.0 | 58.4 | 80.4 | 76.9 | 93.3 | 96.2 | 92.8 | 30.3 | 36.7 | 41.6 | 20.4 | 15.5 | 26.6 | 35.9 | 49.7 | 64.2 |
| CLIP-ViT-B-32 | LAION-2B | C | 2000.0 | 23.9 | 6.6 | 7.7 | 19.5 | 27.9 | 55.2 | 80.9 | 95.1 | 34.2 | 52.4 | 69.0 | 68.4 | 91.8 | 89.9 | 94.8 | 29.4 | 33.8 | 36.8 | 19.5 | 15.0 | 25.8 | 35.0 | 48.4 | 63.1 |
| CLIP-EVA02-B-16 | LAION-2B(1.6B)+ COYO-700M(0.4M) | C | 2000.0 | 16.2 | 3.6 | 4.6 | 12.8 | 19.8 | 40.5 | 65.5 | 88.4 | 20.5 | 29.7 | 40.0 | 44.4 | 69.0 | 71.0 | 83.5 | 18.6 | 19.9 | 23.4 | 17.0 | 8.2 | 16.1 | 23.6 | 34.7 | 49.7 |
| CLIP-ViT-L-14-QG | MetaClip-400M | C | 400.0 | 16.1 | 4.1 | 6.1 | 15.0 | 23.2 | 43.2 | 63.8 | 83.0 | 17.2 | 22.7 | 28.8 | 30.8 | 43.1 | 48.8 | 69.0 | 20.5 | 22.0 | 27.5 | 9.5 | 10.4 | 18.7 | 25.8 | 37.9 | |
| CLIP-ViT-L-14-QG | OPENAI-WIT | C | 400.0 | 17.0 | 3.6 | 5.1 | 12.6 | 25.6 | 43.3 | 60.5 | 78.2 | 18.9 | 25.5 | 32.4 | 36.4 | 43.7 | 46.3 | 68.9 | 16.5 | 22.6 | 28.8 | 11.0 | 8.7 | 15.6 | 21.9 | 32.1 | 45.5 |
| CLIP-ViT-L-14 | DataComp-1B | C | 1400.0 | 13.4 | 2.7 | 3.3 | 10.5 | 11.4 | 26.7 | 51.7 | 77.8 | 7.5 | 22.4 | 28.7 | 29.3 | 42.3 | 44.6 | 67.7 | 16.0 | 17.9 | 20.2 | 7.8 | 7.2 | 13.2 | 18.8 | 28.3 | 40.8 |
| CLIP-ViT-L-14 | MetaClip full CC | C | 2500.0 | 14.5 | 3.5 | 5.3 | 13.8 | 17.4 | 35.6 | 58.0 | 80.9 | 4.4 | 19.6 | 25.8 | 26.8 | 37.0 | 51.3 | 72.7 | 19.0 | 21.0 | 24.0 | 8.3 | 8.5 | 16.1 | 22.0 | 32.4 | 45.5 |
| CLIP-ViT-L-14 | DFN-2B | C | 2000.0 | 10.8 | 2.2 | 3.1 | 9.7 | 18.8 | 35.3 | 55.6 | 77.3 | 11.2 | 18.4 | 27.3 | 28.4 | 47.5 | 53.2 | 79.0 | 16.4 | 17.9 | 21.4 | 9.0 | 6.7 | 13.5 | 19.3 | 28.7 | 41.6 |
| CLIP-EVA02-L-14 | LAION-2B(1.6B)+ COYO-700M(0.4M) | C | 2000.0 | 11.5 | 2.3 | 3.5 | 8.7 | 11.6 | 24.9 | 44.9 | 69.8 | 10.1 | 15.2 | 21.8 | 22.3 | 34.2 | 42.4 | 62.2 | 13.6 | 15.8 | 17.2 | 6.3 | 5.4 | 9.4 | 13.2 | 19.4 | 29.6 |
| CLIP-ViT-H-14 | DFN-5B | C | 5000.0 | 8.7 | 1.9 | 2.7 | 8.7 | 20.2 | 47.9 | 77.7 | 95.0 | 8.9 | 15.3 | 24.2 | 24.1 | 52.6 | 64.8 | 87.4 | 14.4 | 16.2 | 20.9 | 6.9 | 6.4 | 12.2 | 17.5 | 26.0 | 39.3 |
| DINOv2-S | LVD-142M | SS | 142.0 | 7.0 | 1.9 | 2.1 | 8.7 | 20.2 | 47.9 | 77.7 | 95.0 | 13.2 | 17.4 | 26.3 | 26.3 | 50.9 | 53.5 | 90.1 | 17.4 | 21.3 | 26.2 | 40.7 | 9.2 | 17.2 | 24.9 | 37.0 | 53.3 |
| DINOv2-B | LVD-142M | SS | 142.0 | 4.3 | 1.3 | 1.3 | 6.1 | 11.4 | 26.7 | 51.7 | 77.8 | 7.5 | 9.3 | 12.6 | 12.7 | 22.9 | 28.2 | 60.2 | 12.9 | 17.5 | 17.6 | 30.0 | 6.0 | 11.1 | 16.2 | 24.9 | 38.4 |
| DINOv2-L | LVD-142M | SS | 142.0 | 2.5 | 0.9 | 0.8 | 4.0 | 5.7 | 11.6 | 23.8 | 43.6 | 4.4 | 5.2 | 6.4 | 7.0 | 10.6 | 11.5 | 31.1 | 9.4 | 12.0 | 10.7 | 19.4 | 3.6 | 6.7 | 9.2 | 14.2 | 23.4 |
| DINOv2-G | LVD-142M | SS | 142.0 | 2.1 | 0.8 | 0.8 | 3.8 | 4.7 | 8.0 | 14.4 | 26.6 | 3.8 | 4.6 | 5.8 | 6.1 | 8.6 | 9.8 | 18.3 | 8.2 | 13.2 | 10.8 | 17.9 | 3.3 | 6.1 | 7.9 | 11.7 | 19.0 |
| SWIN-L-384 | IN1k | S | 1.2 | 7.2 | 1.2 | 1.4 | 4.7 | 8.2 | 11.7 | 19.5 | 32.5 | 9.6 | 12.3 | 15.8 | 17.6 | 67.6 | 53.6 | 75.3 | 8.6 | 26.8 | 41.5 | 31.2 | 7.6 | 11.1 | 15.5 | 21.3 | 31.5 |
| SWINv2-L-384 | IN21k+IN1k | S | 14.0 | 6.7 | 0.9 | 1.1 | 3.9 | 6.8 | 8.8 | 10.2 | 12.2 | 5.8 | 6.7 | 7.9 | 8.2 | 25.8 | 17.6 | 47.2 | 8.1 | 30.1 | 36.4 | 29.7 | 7.1 | 11.0 | 14.6 | 19.7 | 29.4 |
| ConvNeXt-L | IN21k+IN1k | S | 14.0 | 7.7 | 2.2 | 2.4 | 6.2 | 13.7 | 25.9 | 37.8 | 49.6 | 8.2 | 11.4 | 15.9 | 20.6 | 18.7 | 30.2 | 43.2 | 12.1 | 12.7 | 16.9 | 32.6 | 8.2 | 13.4 | 17.5 | 24.7 | 35.5 |
| YOLO11m-cls | IN1k | S | 1.2 | 16.0 | 4.0 | 4.5 | 15.5 | 50.4 | 90.8 | 98.9 | 99.7 | 31.7 | 58.8 | 82.0 | 91.6 | 97.7 | 91.3 | 98.7 | 21.7 | 30.0 | 38.9 | 57.4 | 25.4 | 39.6 | 51.7 | 65.9 | 77.8 |
| ViT-MAE | IN1k | SS | 1.2 | 33.6 | 8.3 | 13.7 | 30.0 | 68.5 | 94.4 | 99.0 | 99.6 | 58.8 | 81.2 | 92.8 | 94.9 | 98.7 | 98.0 | 99.2 | 46.5 | 54.6 | 80.6 | 76.3 | 30.3 | 45.5 | 56.9 | 69.9 | 81.7 |
| DINOv1-ViT-B-16 | IN1k | SS | 1.2 | 13.2 | 6.7 | 6.7 | 16.1 | 30.1 | 66.4 | 92.2 | 98.9 | 29.5 | 46.7 | 63.0 | 63.2 | 90.6 | 92.4 | 98.9 | 19.5 | 22.8 | 32.2 | 61.6 | 11.7 | 21.9 | 32.7 | 47.4 | 63.8 |
| PaliGemma3b | many | SS | 22256.0 | 11.5 | 2.5 | 3.9 | 12.2 | 26.1 | 59.5 | 84.7 | 96.5 | 17.1 | 26.5 | 38.5 | 46.2 | 68.3 | 69.8 | 91.4 | 16.2 | 19.3 | 23.0 | 15.1 | 11.0 | 20.5 | 29.6 | 42.5 | 57.5 |
| BEiT-B-16-224 | IN21k+IN1k | SS | 14.0 | 10.1 | 1.4 | 1.8 | 6.7 | 12.7 | 29.9 | 56.9 | 83.2 | 13.2 | 21.3 | 31.1 | 34.5 | 67.0 | 71.0 | 73.2 | 16.7 | 17.4 | 27.8 | 36.6 | 6.5 | 11.3 | 15.9 | 23.9 | 37.0 |
| BEiT-L-16-224 | IN21k+IN1k | SS | 14.0 | 6.5 | 1.0 | 1.0 | 4.6 | 8.9 | 20.5 | 41.8 | 71.0 | 8.6 | 12.4 | 17.2 | 17.4 | 42.4 | 63.1 | 69.0 | 11.4 | 13.2 | 21.2 | 26.3 | 4.6 | 7.9 | 10.5 | 15.6 | 25.4 |
| BEiTv2-B-16-224 | IN1k+IN1k | SS | 1.2 | 7.7 | 1.2 | 1.5 | 5.8 | 11.0 | 19.7 | 34.5 | 58.4 | 8.8 | 12.9 | 17.9 | 19.6 | 32.4 | 77.9 | 68.7 | 9.8 | 15.1 | 61.1 | 35.2 | 7.7 | 12.1 | 17.4 | 24.6 | 36.2 |
| BEiTv2-B-16-224 | IN1k+(IN21k+IN1k) | SS | 14.0 | 7.9 | 1.1 | 1.3 | 5.5 | 7.0 | 11.5 | 24.5 | 54.4 | 5.7 | 9.4 | 20.7 | 18.5 | 29.8 | 49.9 | 69.0 | 10.8 | 11.8 | 23.9 | 30.8 | 6.6 | 11.2 | 14.5 | 20.7 | 31.1 |
| SIGLIP-SO400-14-384 | WebLI | C | 22000.0 | 8.5 | 1.8 | 3.3 | 8.9 | 13.0 | 28.1 | 48.7 | 71.7 | 7.9 | 11.8 | 16.5 | 18.9 | 27.9 | 28.8 | 50.4 | 12.0 | 14.7 | 18.2 | 5.1 | 7.7 | 14.4 | 20.9 | 31.3 | 44.7 |
| SIGLIP2-SO400-14-384 | WebLI | C | 22000.0 | 6.3 | 1.4 | 2.0 | 4.5 | 11.8 | 24.0 | 42.0 | 65.9 | 9.3 | 12.1 | 16.0 | 16.2 | 24.6 | 22.0 | 43.1 | 9.8 | 13.4 | 14.7 | 4.0 | 4.3 | 8.7 | 13.8 | 22.3 | 34.6 |
| SIGLIP-B-16-224 | WebLI | C | 22000.0 | 15.3 | 3.8 | 6.5 | 16.1 | 12.8 | 28.0 | 54.7 | 81.8 | 22.2 | 34.1 | 47.7 | 48.1 | 79.1 | 73.7 | 83.1 | 23.9 | 26.0 | 32.0 | 8.0 | 14.0 | 25.8 | 36.0 | 48.8 | 63.1 |
| SIGLIP2-B-16-224 | WebLI | C | 22000.0 | 14.1 | 3.4 | 5.0 | 11.9 | 15.8 | 34.7 | 61.8 | 85.4 | 23.1 | 35.1 | 47.7 | 46.6 | 76.6 | 74.5 | 79.9 | 21.1 | 25.1 | 28.4 | 8.8 | 11.0 | 21.7 | 30.5 | 42.7 | 56.7 |
| Average | | | 3652.0 | 13.1 | 3.1 | 4.2 | 11.5 | 19.8 | 39.1 | 59.6 | 77.7 | 17.9 | 27.4 | 38.0 | 38.6 | 57.4 | 61.4 | 75.5 | 17.8 | 22.0 | 29.1 | 27.1 | 9.7 | 17.1 | 23.6 | 33.7 | 46.8 |