# OpenReview forum: "The Impact of Training Data on Adversarial Robustness"
_NeurIPS.cc/2025/Workshop/Reliable_ML — NeurIPS 2025 - Reliable ML Workshop_

### Official Review · Reviewer_PnmR · 2025-09-21
**The paper raises an interesting question about scaling laws for adversarial robustness but needs broader benchmarks, clearer positioning, and more detail on methodology to strengthen its contribution.**

**Rating:** 6
**Confidence:** 4

**Review:**

Summary: This paper studies how scaling the training data impacts adversarial robustness. The study considers multiple models on the ImageNet dataset and examines how different adversarial attacks affect their performance. Further, a human evaluation is also conducted to establish a benchmark for expected model performance.

Strengths: The work of deriving scaling laws for robustness in vision models addresses a practically relevant and intellectually interesting question.

Weakness/Limitations/Suggestions:
1. Since the focus is on scaling laws, including at least one or two additional benchmark datasets would make the findings more convincing and generalizable.
2. The writing could better position the contribution. For example, in alignment and reasoning research, it is already well established that data quality often outweighs data quantity. Because this paper reaches a similar conclusion, it would be valuable to more rigorously explore the interplay between data quality and performance for vision. For instance, what would the Pareto frontier look like if one could scale either model size or data quality, but not both?
3. The section on approximating the Attack Success Rate is not sufficiently clear. It would help to provide a more detailed explanation of how these approximations are carried out.